# Prefusion-stabilized Hantaan virus glycoprotein nucleic acid vaccine elicits potent neutralizing antibody responses via germinal center activation

Wei Ye [1,7] ✉, Yamei Dang[1,7], Yuan Wang [1,7], Qiqi Yang[1,7], Hui Zhang [1,7], Chuantao Ye [2,7], Jing Wei [1,3,7], Jiawei Pei[1], Xuemin Pei[1,4], Dongshen Jiang[1,5], Xiaojing Yang[1,5], Xiaolei Jin[6], Hongwei Ma[1], He Liu[1], Liang Zhang[1], Linfeng Cheng[1], Yangchao Dong[1], Yingfeng Lei [1] ✉, Zhikai Xu [1] ✉ & Fanglin Zhang [1] ✉

Old World *orthohantaviruses*, including Hantaan virus (HTNV), cause hemorrhagic fever with renal syndrome (HFRS) in Eurasia. Available inactivated vaccines often induce low neutralizing antibodies and short-term protection. We evaluated nucleic acid vaccines expressing a prefusion-stabilized HTNV glycoprotein in female BALB/c mice. Both DNA and mRNA-LNP versions elicited robust neutralizing antibodies by strongly activating germinal centers, which protected mice against high-dose HTNV challenge. We further tested heterologous prime-boost regimens, where mice primed with inactivated vaccine received different boosters. All boosters increased neutralizing titers, but only the prefusion-stabilized glycoprotein mRNA-LNP vaccine raised titers to the level achieved by its own full primary vaccination course. This demonstrates the immunogen's superiority in developing next-generation vaccines and its unique ability to potently recall memory B cells induced by suboptimal inactivated vaccines. Thus, prefusion-stabilized glycoprotein-based nucleic acid vaccines are promising candidates for advanced *orthohantavirus* vaccine development.

Rodent-borne *Orthohantavirus* (hereafter referred to as hantaviruses) causes two severe, often fatal human diseases; their distribution aligns with that of their natural rodent reservoirs[1-3]. In Eurasia, "Old World" hantaviruses (OWHs) such as Hantaan virus (HTNV), Dobrava-Belgrade virus ( DOBV ) , and Puumala virus (PUUV) cause hemorrhagic fever with renal syndrome (HFRS), which has a mortality rate of up to 15%[3-5]. The Seoul virus (SEOV) is distributed worldwide along with its natural host *Rattus norvegicus*[3-5]. In the Americas, "New World" hantaviruses (NWHs) such as Sin Nombre virus and Andes virus (ANDV) cause hantavirus pulmonary syndrome (also called hantavirus cardio-pulmonary syndrome), with mortality rates reaching up to 45%[2,3]. Climate change and urbanization are altering rodent host distributions, leading to unexpected hantavirus outbreaks[6,7]. In East Asia, an inactivated HTNV vaccine derived from mouse brain (Hantavax™) is licensed

[1]Department of Microbiology, School of Basic Medicine, Airforce Medical University: Fourth Military Medical University, Xi'an, Shaanxi, China. [2]Department of Infectious Diseases, Tangdu Hospital, Airforce Medical University: Fourth Military Medical University, Xi'an, Shaanxi, China. [3]Center for Disease Control and Prevention of Shaanxi Province, Xi'an, Shaanxi, China. [4]School of Medicine, Northwest University, Xi'an, Shaanxi, China. [5]School of Medicine, Yan'an University, Yan'an, Shaanxi, China. [6]Student Brigade, School of Basic Medicine, Airforce Medical University: Fourth Military Medical University, Xi'an, Shaanxi, China. [7]These authors contributed equally: Wei Ye, Yamei Dang, Yuan Wang, Qiqi Yang, Hui Zhang, Chuantao Ye, Jing Wei. ✉e-mail: virologyyw@fmmu.edu.cn; yflei@fmmu.edu.cn; zhikaixu@fmmu.edu.cn; flzhang@fmmu.edu.cn

in South Korea, and several bivalent inactivated vaccines against HTNV and SEOV are used in China[8]. Field investigations have shown that while these inactivated vaccines can induce a binding antibody response, the resulting neutralizing antibody (NAb) titers are often low and transient[8]. Therefore, designing immunogens capable of eliciting high-titer, durable NAb responses remains a critical need for developing effective prophylactic vaccines against hantaviruses.

Hantaviruses, belonging to the family *Hantaviridae* and genus *Orthohantavirus*, are enveloped viruses with a tri-segmented negative-sense RNA genome[1,3,9]. The genome encodes an RNA polymerase, a nucleocapsid protein (NP), and a glycoprotein (GP) precursor that is proteolytically cleaved into N-terminal Gn and C-terminal Gc subunits[1,3,9]. On the virion surface, Gn and Gc form a metastable heterotetrameric spike[9–14]. Gn mediates cellular receptor attachment and shields cryptic epitopes on Gc, which is a class II fusion protein that drives membrane fusion during endocytosis[9–14]. Both Gn and Gc are targets of NAbs, driving vaccine development efforts focused on these proteins individually or in combination[15–20]. Recent advances in structural biology have resolved the architecture of the hantavirus GP, including the Gn head (GnH) and the Gc ectodomain[11,12,21,22]. These structures provide a mechanistic basis for neutralization by both previously known and newly isolated NAbs[23–28].

Despite these efforts, the rational design of vaccine development for hantaviruses has lagged compared to that for other viruses, such as coronaviruses and respiratory syncytial virus (RSV)[29,30]. Building on recent structural insights, we engineered an HTNV GP immunogen by introducing three cysteine mutations to stabilize its prefusion conformation and prevent structural rearrangements. We then evaluated this engineered GP delivered via both a DNA vaccine platform and a lipid nanoparticle (LNP)-encapsulated mRNA vaccine. This work provides proof-of-concept for a rational, structure-based antigen design strategy in developing an efficacious hantavirus vaccine. These findings support the further investigation and development of this strategy.

## Results

### Prefusion-stabilized HTNV GP DNA vaccination elicits a significant NAb response

The HTNV GP contains multiple endogenous disulfide bonds[31]. As the HTNV Gc is a class II membrane fusion protein, we hypothesized that engineering additional disulfide bonds would increase the energy barrier for the conformational change required for membrane fusion.

First, we introduced an inter-protomer disulfide bond within Gc by replacing glycine 835 with cysteine (G835C; numbering corresponds to the GP precursor) to increase Gn/Gc spike stability, designating this mutant as GP-C1 (Fig. 1a, Supplementary Fig. 1a, b)[13]. Because the GnH is loosely attached to Gc, we next introduced an inter-subunit disulfide bond by engineering serine 291 and threonine 731 to cysteine (S291C/T731C). This design was based on structural modeling of the HTNV GP using the prefusion glycoprotein complex of ANDV (PDB: 6Y5F)[12]; the resulting mutant was designated GP-C2 (Fig. 1a, b, Supplementary Fig. 1a). Finally, we constructed a triple mutant combining all three mutations (S291C/T731C/G835C), designated GP-C3 (Fig. 1a, Supplementary Fig. 1a).

To determine whether these mutations stabilized the prefusion state, we expressed C-terminal Myc-tagged proteins and analyzed their migration under reducing and non-reducing SDS-PAGE conditions, alongside a WAASR mutant, which produces an uncleaved GP precursor (Fig. 1c)[9].

Under reducing conditions, wild-type (WT) GP migrated primarily as a ~50 kDa monomer (representing Gc), while the WAASR mutant appeared as an uncleaved ~120 kDa precursor (Fig. 1c). In contrast, GP-C2 and GP-C3 showed a prominent high-molecular-weight band (>480 kDa) (Fig. 1c), indicating the preservation of Gn/Gc heterotetramers despite β-mercaptoethanol (β-ME) treatment. Under non-

reducing conditions, while all constructs showed some tetrameric form, the monomeric Gc band was nearly absent in GP-C3 and substantially reduced in GP-C2 compared to WT (Fig. 1c). When cloned into the pVAX1 DNA vaccine vector, GP-C1, GP-C2, and GP-C3 maintained their stabilized conformation, with GP-C3 showing the strongest effect (Supplementary Fig. 1c).

We next performed a syncytium formation assay to evaluate functional stabilization. In HEK-293 T cells (chosen for their high transfection efficiency at 24 h; Supplementary Fig. 1d), the GP-C2 and GP-C3 mutants produced fewer syncytia than WT, with GP-C3 yielding the fewest GFP-positive syncytia (Fig. 1d, Supplementary Fig. 1e). These data confirm that the engineered cysteine mutations stabilize the prefusion conformation of HTNV GP.

We then evaluated the immunogenicity of these GP mutants in BALB/c mice, using the inactivated vaccine as a positive control (Fig. 1e). Mice received three vaccinations at 4-week intervals, with serum collected 2 weeks after each dose. Throughout the study, all vaccinated mice remained active and healthy, with no clinical signs of distress or systemic toxicity observed, indicating the favorable safety profile of the vaccine candidates. All GP immunogens induced robust humoral responses, with GP-C3 performing best (Fig. 1f, g). While GP-specific IgG levels at week 2 were similar between GP-C3 and WT, the GP-C3 group already exhibited significantly higher NAb titers against HTNV than the WT and inactivated vaccine groups (Fig. 1g). Antibody titers peaked at day 42 in all groups, with GP-C3 again inducing the highest levels of both GP-specific IgG and NAbs. Notably, by day 70, the NAb titer in the inactivated vaccine group had declined by half, whereas both NAb and GP-specific antibody titers in the GP-C3 group remained at their peak levels, suggesting robust antibody affinity maturation (Fig. 1f, g).

We also assessed cross-neutralizing activity against SEOV, a clade II hantavirus with considerable sequence similarity to HTNV[25]. At both 14 and 70 days post-immunization, GP-C3 induced the highest SEOV NAb titers among all groups (Fig. 1h, Supplementary Fig. 1f). Although GP-C2 elicited strong HTNV-specific responses at day 14, its SEOV NAb titer was lower than those induced by GP-C1 and WT (the difference was not significant), but increased by day 70 (Fig. 1h, Supplementary Fig. 1f). Collectively, these results indicate that the disulfide-bond-stabilized GP-C3 is an ideal immunogen, inducing potent homologous and cross-neutralizing antibodies.

Finally, we investigated the T-cell responses induced by the DNA vaccines. Splenocytes were harvested 70 days post-immunization (Fig. 1e). Upon in vitro stimulation with HTNV-GP peptides, GP-C1, GP-C2, and GP-C3 induced stronger IFN-γ and IL-2 recall responses than the inactivated vaccine, with GP-C3 eliciting the highest spot-forming cells (SFCs) levels (Fig. 1i, j). Correspondingly, flow cytometry analysis showed that the percentages of IFN-γ-secreting CD4$^+$ and CD8$^+$ T cells were significantly higher in the GP-C3 group than in all other groups (Fig. 1k, l, Supplementary Fig. 2a, b). A similar pattern was observed for TNF-α-secreting T cells (Supplementary Fig. 2c, d). In contrast, Th2-associated cytokine responses (IL-4 and IL-10 SFCs) were similar across all DNA vaccine groups and were not significantly improved over the inactivated vaccine (Fig. 1m, n). Flow cytometry analysis of IL-4-secreting CD4$^+$ and CD8$^+$ T cells yielded consistent results (Fig. 1o, p, Supplementary Fig. 2e, f). Together, these data demonstrate that the prefusion-stabilized HTNV GP DNA vaccine, particularly the GP-C3 construct, elicits a potent Th1-skewed cellular immune response.

### Prefusion-stabilized HTNV GP DNA vaccine confers potent protection against viral challenge and tissue injury

To determine whether prefusion-stabilized HTNV GP DNA vaccination confers protective immunity, mice were challenged with $5 \times 10^5$ focus-forming units (FFUs) of HTNV strain 76–118 via intramuscular (i.m.) injection two weeks after the final immunization. All animals were

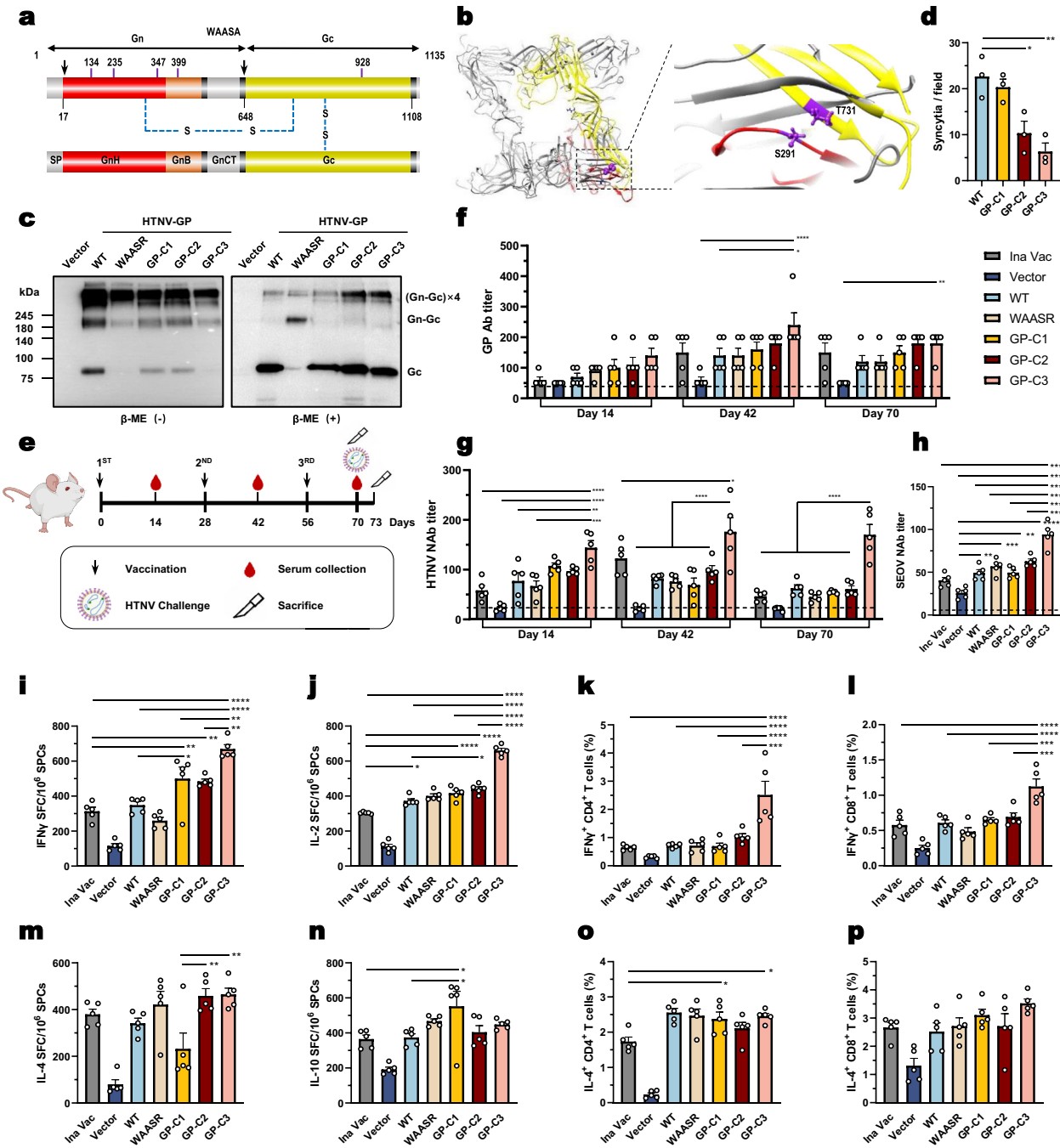

euthanized three days post-infection, and viral RNA loads in the lung, liver, kidney, and spleen were quantified by qRT-PCR.

The viral RNA level was markedly reduced in all immunogen-vaccinated groups compared to the pVAX1 vector control group (Fig. 2a, b, Supplementary Fig. 3a, b). In the lung (Fig. 2a), GP-C3 immunization reduced viral RNA to 9.24% of the vector control level, corresponding to a 10.8-fold reduction. This protective efficacy surpassed that of the inactivated vaccine (1.7-fold reduction) and the wild-type (WT) GP vaccine (1.6-fold reduction). The GP-C1 and GP-C2 constructs also conferred strong protection, with 6.5-fold and 5.4-fold reductions in lung viral RNA, respectively. A similar trend was observed in other tissues. In the liver (Fig. 2b), GP-C3 immunization led to a 7.2-fold reduction in viral RNA. Notably, although GP-C1-induced antibody titers were not significantly higher than those elicited by the inactivated vaccine or WT GP, it still provided substantial protection,

reducing viral RNA in the lung, liver, and spleen by 6.5-, 5.2-, and 6.1-fold, respectively, suggesting protective mechanisms beyond peak antibody titers.

Consistent with the viral load data, immunofluorescence staining for HTNV nucleocapsid protein (NP) revealed substantially less antigen in the lung, kidney, and spleen tissues of GP-C3-immunized mice compared to the pVAX1, inactivated vaccine, and WT GP groups (Fig. 2c, Supplementary Fig. 3c–g).

Hematological and serological analyses further demonstrated the protective efficacy of the vaccines. Total white blood cell (WBC) and lymphocyte (LYM) counts in all immunogen-vaccinated groups were maintained close to the normal range for adult BALB/c mice and were significantly lower than the elevated levels in the pVAX1 control group (Fig. 2d, e). Neutrophil (NEU) counts were comparable across all vaccinated groups (Supplementary Fig. 3h). Assessment of liver and

**Fig. 1 | Design, in vitro characterization, and immunogenicity of prefusion-stabilized HTNV GP DNA vaccines. a** Schematic of the HTNV glycoprotein precursor (GPC). Domains are labeled: Gn and Gc ranges, the signal peptide cleavage site (arrow), N-linked glycosylation sites (violet), the transmembrane domain (black), and engineered disulfide bonds (blue dashes). Ectodomains are colored: Gn head (GnH, red), Gn base (GnB, orange), and Gc (yellow). **b** Homology model of the HTNV envelope glycoprotein prefusion complex (Gn/Gc)4, based on ANDV (PDB: 6Y5F). GnH is shown in red and Gc in yellow. Engineered disulfide bond residues (GnH S291 and Gc T731) are highlighted in violet. **c** HEK-293T cells were transfected with pCAGGS vector or Myc-tagged plasmids expressing WT, WAASR, GP-C1, GP-C2, or GP-C3 constructs. After 24 h, lysates were prepared using RIPA buffer, immunoprecipitated with anti-Myc magnetic beads, and analyzed by Native PAGE (left) or reducing SDS-PAGE (right) for immunoblotting with an anti-Myc antibody (β-ME: β-mercaptoethanol). **d** Quantification of syncytia per field 24 h post-transfection with different HTNV GP plasmids, assessed by inverted fluorescence microscopy. Data are mean ± SEM, $n = 3$ biological replicates. Significance was determined by one-way ANOVA with Dunnett's multiple comparisons test. *$P < 0.05$, **$P < 0.01$. **e** Immunization and challenge scheme. BALB/c mice (6-8 weeks old) received three intramuscular (i.m.) immunizations with a DNA vaccine or inactivated vaccine. Sera/tissues were collected at the indicated times. For challenge, mice were i.m. administered $5 \times 10^5$ FFU of HTNV strain 76–118 and sacrificed 3 days later. **f, g** HTNV GP-specific IgG titers (**f**) and HTNV-neutralizing antibody (NAb) titers (**g**), determined by ELIZA and $FRNT_{50}$, respectively, at days 14, 42, and 70 post-immunization. The dashed line indicates the detection limit. Data are mean ± SEM, $n = 5$ mice per group. Significance was determined by two-way ANOVA with Tukey's test, a two-sided post-hoc test adjusted for multiple comparisons. *$P < 0.05$, **$P < 0.01$, ***$P < 0.001$, ****$P < 0.0001$. **h** Seoul virus (SEOV)-NAb titers in serum at day 70 post-immunization. **i, j** IFN-γ (**i**) and IL-2 (**j**) secretion by splenocytes in response to HTNV-GP peptide stimulation (10 µg/mL), measured by ELISpot and expressed as spot-forming units (SFU) per million splenocytes (day 70). **k, l** Frequency of IFN-γ⁺ CD4⁺ (**k**) and CD8⁺ T cells (**l**), determined by intracellular cytokine staining after peptide stimulation. **m, n** IL-4 (**m**) and IL-10 (**n**) secretion by splenocytes (ELISpot). **o, p** Frequency of IL-4⁺ CD4⁺ (**o**) and CD8⁺ T cells (**p**) (intracellular staining). **h–p** Data are mean ± SEM, $n = 5$ mice per group. Significance was determined by one-way ANOVA with Tukey's test. *$P < 0.05$, **$P < 0.01$, ***$P < 0.001$. Exact $P$-values are provided in the Source Data file. Source data are provided as a Source Data file.

kidney function showed that the GP-C3 group had the lowest levels of aspartate aminotransferase (AST; Fig. 2f) and creatinine (CREA; Fig. 2g). Alanine aminotransferase (ALT), albumin (ALB), and blood urea nitrogen (BUN) levels in all immunogen groups were also within the normal range (Supplementary Fig. 3i–k), indicating effective protection against HTNV-induced hepatic and renal injury.

Histopathological analysis of tissue sections from challenged mice provided direct evidence of protection (Fig. 2h, i, Supplementary Fig. 3l, m). Lungs of mice immunized with the pVAX1 (Vector) exhibited discernible pathology, including thickened alveolar septa, whereas lung morphology in the GP-C3 group remained largely normal (Fig. 2h). The pVAX1 group also displayed kidney injury, manifested as renal tubulointerstitial hemorrhage and erythrocyte accumulation within glomerular capillary lumina with luminal obstruction; as well as liver damage characterized by hepatocellular edema, and splenic inflammation. In contrast, these pathological alterations were markedly attenuated in the GP-C3 group, which showed no discernible pathological alterations in any of these tissues.

Collectively, these results demonstrate that the prefusion-stabilized HTNV GP DNA vaccine, particularly the GP-C3 construct, provides robust and comprehensive protection against HTNV challenge, effectively suppressing viral replication and preventing virus-induced tissue injury.

## Prefusion-stabilized HTNV GP DNA vaccine elicits prolonged and durable NAb response

Given the protective efficacy of the prefusion-stabilized vaccine, we next evaluated the longevity of the humoral immune response it elicits. Mice received three immunizations at 4-week intervals, and serum and splenocytes were analyzed at day 180 for long-term immune evaluation (Fig. 3a).

HTNV GP-specific IgG antibody titers in the GP-C3 group remained significantly higher than those in the WT and inactivated vaccine groups at this late time point (Fig. 3b). Notably, NAb titers in the GP-C3 group showed no evident decline, maintaining high levels at day 180 (144.91 ± 40.06) compared to day 70 (172.81 ± 33.99; see also Fig. 1h), indicating a remarkably durable neutralizing antibody response (Fig. 3c).

A well-coordinated immune response often involves an initial pro-inflammatory Th1 phase followed by a shift toward a Th2 profile, which supports antibody affinity maturation and memory B cell (MBC) formation[32]. To assess T-cell immunity over time, we performed ELISpot assays for IFN-γ and IL-4 recall responses at day 180. Only the inactivated vaccine and the GP-C3 construct maintained substantial levels of IFN-γ-secreting T cells, whereas responses in the other DNA vaccine groups diminished considerably by 6 months (Fig. 3d). Flow cytometry analysis corroborated these findings, showing no significant difference in the frequencies of IFN-γ- and TNF-α-secreting CD4⁺ and CD8⁺ T cells between the GP-C3 and inactivated vaccine groups (Fig. 3e, f, Supplementary Fig. 4a–d).

In contrast, a clear Th2-skewed response emerged in the long term. ELISpot analysis revealed that GP-C1, GP-C2, and GP-C3 all elicited high levels of IL-4-secreting cells at day 180 (Fig. 3g). Specifically, the GP-C3 group exhibited the highest frequencies of IL-4-secreting CD4⁺ and CD8⁺ T cells (Fig. 3h, i, Supplementary Fig. 4e, f), indicating a robust and sustained Th2-type response.

Together, these results demonstrate that the GP-C3 immunogen induces high and durable NAb titers, which are maintained for at least 6 months. This sustained humoral immunity is associated with a coordinated immune shift, where an early Th1 response transitions into a dominant and persistent Th2-skewed profile over time.

## Prefusion-stabilized HTNV GP DNA vaccination elicits robust primary germinal center responses

Germinal center (GC) formation, driven by T follicular helper (Tfh) cells, is essential for B cell affinity maturation and the generation of high-affinity antibodies[33–35]. Given that the prefusion-stabilized GP-C3 immunogen induced potent and durable NAb responses, we hypothesized that it would efficiently promote Tfh cell differentiation and GC formation.

To test this, BALB/c mice received a single i.m. dose of the different vaccines. Draining lymph nodes (LNs) were analyzed 7 days post-immunization—a well-established peak for germinal center formation—by flow cytometry seven days post-immunization. The frequency of Tfh cells (CD4⁺ B220⁻ PD1⁺ CXCR5⁺) in the LNs of GP-C3-immunized mice was significantly higher than in mice immunized with the empty vector, inactivated vaccine, or WT GP (Fig. 4a–c, Supplementary Fig. 5a).

Consistently, GP-C3 vaccination also induced a significantly higher frequency of GC B cells (CD19⁺ CD38⁻ CD95⁺) compared to the control groups (Fig. 4d, e, Supplementary Fig. 5b).

We next performed histological immunofluorescence staining to visualize GC structures directly. GCs were identified as GL7⁺ B cell clusters within B220⁺ follicles and distinct from adjacent CD3⁺ T cell zones. The number of detectable GCs was significantly greater in the GP-C3 group than in all other groups (Fig. 4f, g).

These data demonstrate that a single dose of the prefusion-stabilized GP-C3 immunogen potently initiates a germinal center reaction, characterized by robust Tfh cell expansion and GC B cell

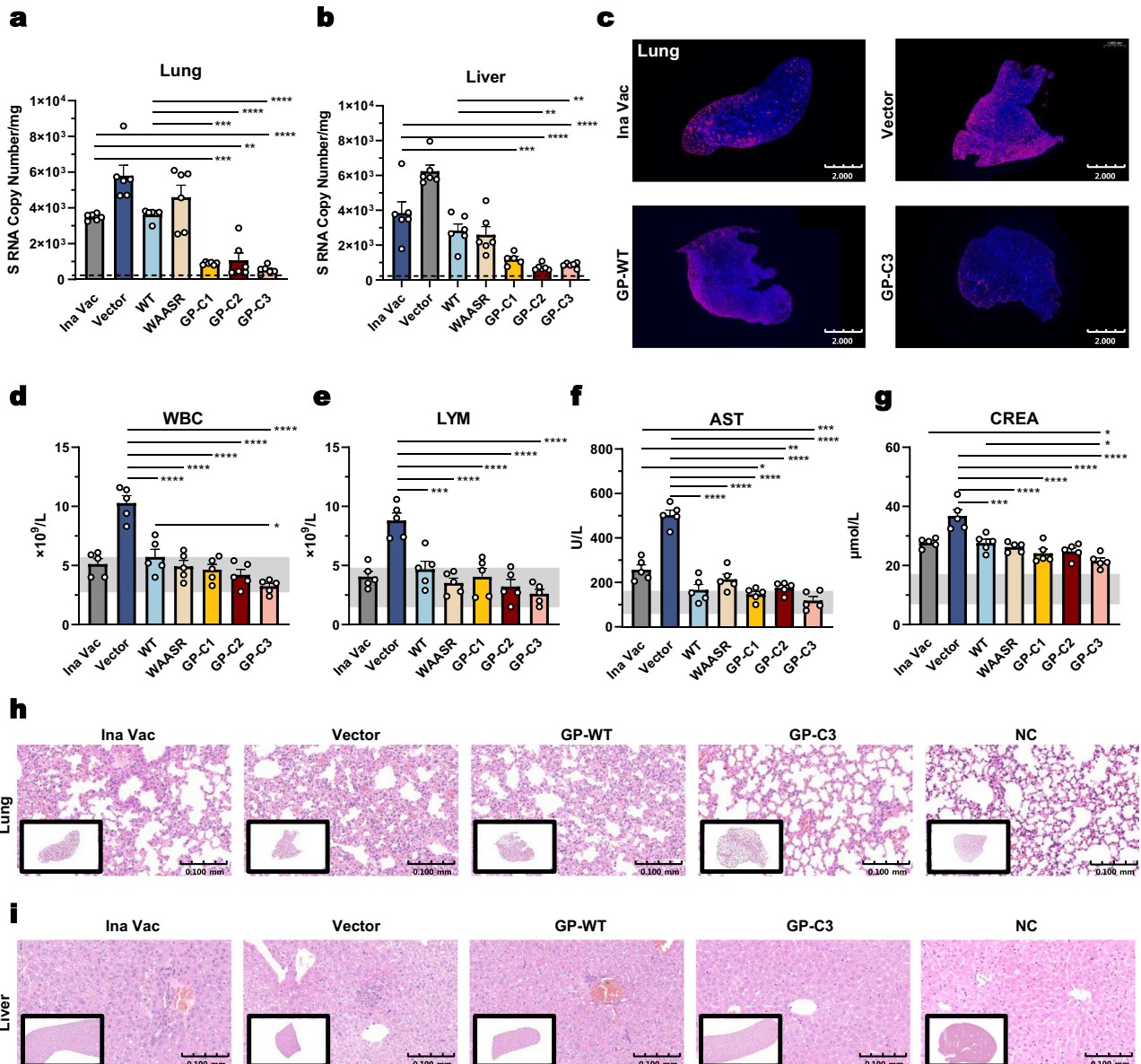

**Fig. 2 | Protective efficacy of prefusion-stabilized HTNV GP DNA vaccine against viral challenge. a, b** Viral RNA loads (copies/mg) in lung (**a**) and liver (**b**) tissues were quantified by qRT-PCR. The dashed line indicates the detection limit. **c** Representative immunofluorescence images of lung sections showing HTNV nucleoprotein (NP, red) and nuclei (DAPI, blue). Scale bar, 2 mm. **d–g** Analysis of whole blood counts and serum biochemistry at 3 days post-challenge: white blood cells (WBC, **d**), lymphocytes (LYM, e), aspartate aminotransferase (AST, **f**), and creatinine (CREA, **g**). **h**, **i** Representative hematoxylin and eosin (H&E)-stained sections of lung (**h**) and liver (**i**) at 3 days post-infection. Scale bar, 100 µm. Data are presented as mean ± SEM. For viral RNA loads (**a**, **b**), $n = 6$ mice per group; for blood counts and biochemistry (**d–g**), $n = 5$ mice per group. For 2c, h–i, this experiment was independently repeated three times with similar results. Statistical significance was determined by one-way ANOVA with Tukey's multiple comparisons test. $*P < 0.05$, $**P < 0.01$, $***P < 0.001$, $****P < 0.0001$. Exact $P$ values are provided in the Source Data file. Source data are provided as a Source Data file.

accumulation, providing a cellular basis for its superior antibody responses.

## mRNA-LNP delivery of prefusion-stabilized HTNV GP elicits potent humoral and cellular immunity

Having established GP-C3 as the lead immunogen in the DNA vaccine platform, we next evaluated its performance in a clinically relevant mRNA-LNP platform. The mRNA was synthesized with a canonical Cap1 structure and incorporated N1-methylpseudourine (m1Ψ) nucleoside modifications to enhance stability and translational efficiency while reducing innate immunogenicity (Fig. 4a, Supplementary Fig. 6a).

Both GP-WT and GP-C3 mRNAs were successfully encapsulated into LNPs with properties suitable for immunization (Fig. 5a, b, Supplementary Fig. 6b, c). In vitro, these mRNA-LNPs mediated efficient expression of HTNV GP in Huh7 cells, as detected by Gc protein, while empty LNPs showed no signal (Fig. 5c). To evaluate immunogenicity, mice received two i.m. doses (2 or 10 µg) of GP-WT or GP-C3 mRNA-LNP at a 3-week interval, with empty LNPs and inactivated vaccine serving as controls (Fig. 5d). As observed in the DNA vaccine regime, no clinical signs of distress or systemic toxicity were observed, indicating a similarly favorable safety profile of the mRNA-LNP platform.

The mRNA-LNP vaccines, particularly at the 10 µg dose, elicited potent antibody responses. Serum analysis revealed that the 10 µg GP-

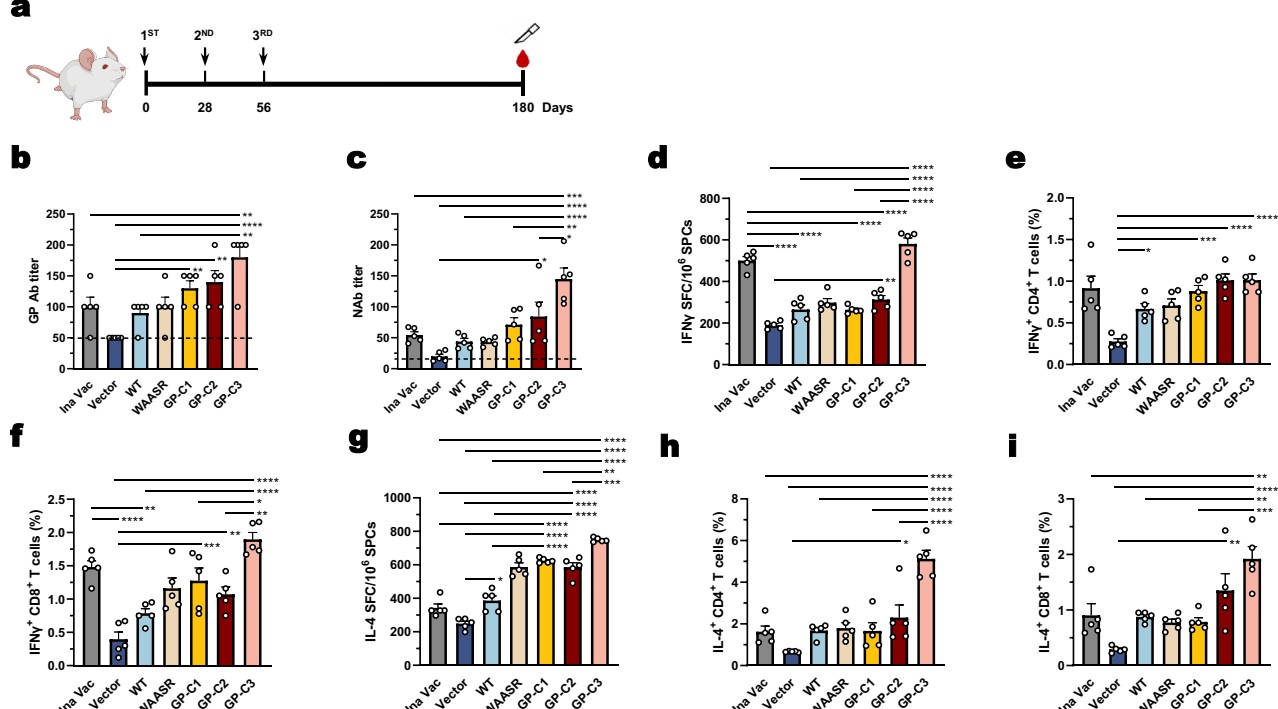

**Fig. 3 | Prefusion-stabilized HTNV GP DNA vaccine induces durable humoral and cellular immunity. a** Schematic of the long-term immunization schedule. Mice received three vaccine doses at 4-week intervals and were sacrificed at day 180. **b, c** Humoral immune responses at endpoint: HTNV GP-specific antibody titers (**b**) and HTNV-neutralizing antibody (NAb) titers (**c**). The dashed line indicates the detection limit. **d–i** Cellular immune responses in splenocytes at day 180: frequencies of IFN-γ-secreting cells (**d**), IFN-γ⁺ CD4⁺ T cells (**e**), IFN-γ⁺ CD8⁺ T cells (**f**), IL-4-secreting cells (**g**), IL-4⁺ CD4⁺ T cells (**h**), and IL-4⁺ CD8⁺ T cells (**i**) in response to HTNV-GP peptide stimulation, as determined by ELISpot or flow cytometry. Data are presented as mean ± SEM, $n = 5$ mice per group. Statistical significance was determined by one-way ANOVA with Tukey's multiple comparisons test. *$P < 0.05$, **$P < 0.01$, ***$P < 0.001$, ****$P < 0.0001$. Exact $P$-values are provided in the Source Data file. Source data are provided as a Source Data file.

C3 group produced significantly higher HTNV GP-specific IgG titers than all other groups (Fig. 5e). Crucially, this group also achieved the highest neutralizing antibody (NAb) titers against HTNV (Fig. 5f).

Interestingly, despite the self-adjuvanting properties of mRNA-LNPs, the NAb levels induced by GP-WT and 2 µg GP-C3 mRNA-LNP were comparable to, rather than substantially exceeding, those of the inactivated vaccine. This underscores the intrinsic potency of the prefusion-stabilized GP-C3 immunogen across delivery platforms. Furthermore, the GP-C3 mRNA-LNP vaccine elicited strong cross-neutralizing activity against SEOV, with titers reaching a level comparable to the homologous HTNV response (Fig. 5g). A direct comparison between vaccine platforms, accounting for their respective bleeding endpoints, confirmed that the mRNA-LNP formulation achieved higher peak NAb titers than the DNA vaccine (HTNV: 247.16 vs. 170.12; SEOV: 242.64 vs. 94.18), as summarized in Supplementary Table 1.

We next evaluated cellular immunity. Splenocytes harvested four weeks post-immunization were stimulated with HTNV GP peptide pools. Consistent with the DNA vaccine results, the 10 µg GP-C3 mRNA-LNP group mounted the strongest IFN-γ and IL-2 recall responses among all groups (Fig. 5h, i). Flow cytometry confirmed significantly higher frequency of IFN-γ-secreting CD4⁺ and CD8⁺ T cells in this group (Fig. 5j, k, Supplementary Fig. 6d, e). In contrast, IL-4 responses across mRNA-vaccinated groups were similar to those observed with the DNA vaccine platform, with GP-C3 showing no marked enhancement (Fig. 5l–n, Supplementary Fig. 6f, g). Additionally, IL-10 levels in the GP-WT groups were lower than in the inactivated vaccine group (Fig. 5o).

In summary, the prefusion-stabilized GP-C3 immunogen delivered via mRNA-LNP elicits robust and balanced immune responses, characterized by high-titer, cross-reactive NAbs and a Th1-skewed cellular profile, confirming its efficacy across nucleic acid vaccine platforms.

## mRNA-LNP delivery of prefusion-stabilized HTNV GP provides optimal protection against HTNV infection

To evaluate protective efficacy, vaccinated mice were challenged with $5 \times 10^5$ FFUs of HTNV at four weeks post-immunization and analyzed three days later (Fig. 5d). qRT-PCR analysis showed markedly reduced HTNV RNA levels in the lungs, liver, and kidneys of all immunogen-vaccinated groups compared to the LNP placebo group (Fig. 6a, b, Supplementary Fig. 7a, b). Notably, the GP-C3 mRNA-LNP group exhibited the lowest viral burden, with lung RNA levels significantly lower than those in the GP-WT group and approaching sterile protection (Fig. 6a). Consistent with this, immunofluorescence staining for HTNV NP antigen revealed minimal viral protein in the lung, kidney, and spleen tissues of GP-C3-vaccinated mice (Fig. 6c, Supplementary Fig. 7c–g). These results indicate that GP-C3 mRNA-LNP immunization effectively restricts viral replication and dissemination.

Analysis of hematological and serological parameters further demonstrated protection from virus-induced pathology. White blood cell (WBC), lymphocyte (LYM), and neutrophil (NEU) counts in the GP-C3 group were significantly lower than in the LNP placebo group but remained within the normal range for healthy mice (Fig. 6d–f). Levels of the liver injury markers alanine aminotransferase (ALT) and aspartate aminotransferase (AST) were also lower in all immunogen-vaccinated groups compared to the LNP placebo (Fig. 6g, Supplementary Fig. 7h), suggesting protection against virus-induced liver

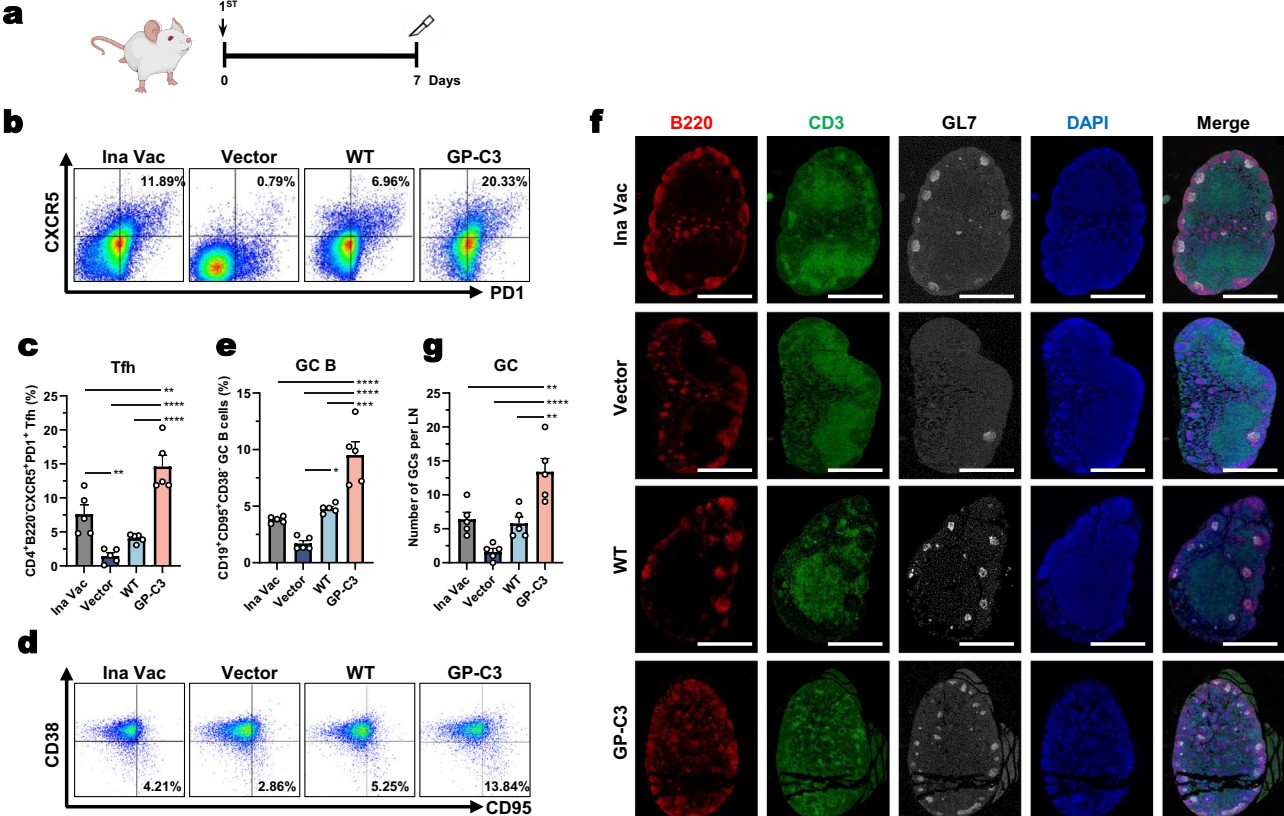

**Fig. 4 | Prefusion-stabilized HTNV GP DNA vaccine elicits robust germinal center (GC) responses. a** Immunization schedule. Inguinal lymph nodes (LNs) were collected from 6- to 8-week-old female BALB/c mice at 7 days post-immunization. **b**, **c** Analysis of T follicular helper (Tfh) cells: representative flow cytometry plots (**b**) and quantification of the percentage of Tfh cells in draining LNs (**c**). **d**, **e** Analysis of germinal center B (GC B) cells: representative flow cytometry plots (**d**) and quantification of the percentage of GC B cells (**e**). **f**, **g** Immunofluorescence analysis of germinal centers in LN sections stained for B220 (red), CD3 (green), and GL7 (white) (**f**). Scale bar, 500 μm. Quantification of the average number of GCs per group is shown in (**g**). Data are presented as mean ± SEM, $n = 5$ mice per group. Statistical significance was determined by one-way ANOVA with Tukey's multiple comparisons test. **$P < 0.01$, ***$P < 0.001$, ****$P < 0.0001$. Exact $P$-values are provided in the Source Data file. Source data are provided as a Source Data file.

injury. Furthermore, the GP-C3 group maintained near-normal levels of renal function markers, including albumin (ALB), blood urea nitrogen (BUN), and creatinine (CREA) (Fig. 6h, Supplementary Fig. 7i, j), suggesting effective protection against HTNV-induced kidney damage.

Histopathological examination provided morphological evidence of protection. Lung sections from the LNP placebo group showed discernible pathology, including thickened alveolar septa (Fig. 6i). In contrast, all immunogen-vaccinated groups displayed attenuated lung injury to varying degrees, with the GP-C3 group exhibiting the least severe alveolar structural disruption (Fig. 6i). Similarly, examination of liver, kidney, and spleen sections revealed that GP-C3-immunized mice developed only mild pathological alterations, with no typical signs of inflammatory infiltration or notable edema observed (Fig. 6j, Supplementary Fig. 7k,l).

In summary, the prefusion-stabilized GP-C3 immunogen delivered via mRNA-LNP confers optimal protection against high-dose HTNV challenge, effectively suppressing viral replication, preserving critical organ function, and preventing acute tissue injury.

**mRNA-LNP delivery of prefusion-stabilized HTNV GP elicits robust germinal center responses**

Since the GP-C3 mRNA-LNP vaccine induced robust NAb responses and protection, we hypothesized that it could also efficiently trigger primary GC reactions. To test this, inguinal LNs were collected from mice 7 days post-vaccination for analysis. Flow cytometry revealed that Tfh cells constituted nearly one-third of the CD4$^+$ T cell population in the GP-C3 group, significantly higher than in the GP-WT and inactivated vaccine groups (Fig. 7a, b), indicating a potent GC response. GC B cells stimulated by Tfh and follicular dendritic cells rapidly proliferate and undergo antibody diversification, differentiating into MBCs and long-lived plasma cells (LLPCs)[33–35]. Consistently, the frequency of CD19$^+$ CD38$^-$ GL7$^+$ CD95$^+$ GC B cells was also significantly elevated in the GP-C3 group compared to these controls (Fig. 7c, d). Histological immunofluorescence analysis of LNs confirmed these findings, showing that both the number and area of GCs were greatest in the GP-C3 group (Fig. 7e, f). These results demonstrate that the prefusion-stabilized GP-C3 mRNA-LNP vaccine is highly effective at inducing a strong primary germinal center response.

**mRNA-LNP delivery of prefusion-stabilized HTNV GP effectively boosts immunity primed by inactivated vaccine**

Despite decades of field usage in endemic areas, epidemiological investigations have revealed comparably low and transient levels of HFRS-inactivated vaccine-induced NAbs[36,37]. We therefore investigated whether a heterologous boost with GP-C3 nucleic acid vaccines could recall and amplify this pre-existing immunity. Mice were primed with two doses of inactivated vaccine and, 180 days later, received a booster immunization with different vaccine regimens, with phosphate-buffered saline (PBS) serving as a naïve placebo for all three doses (Fig. 8a). Two weeks after the final booster dose, the mice were sacrificed, and both blood and spleen samples were collected for immunological assays.

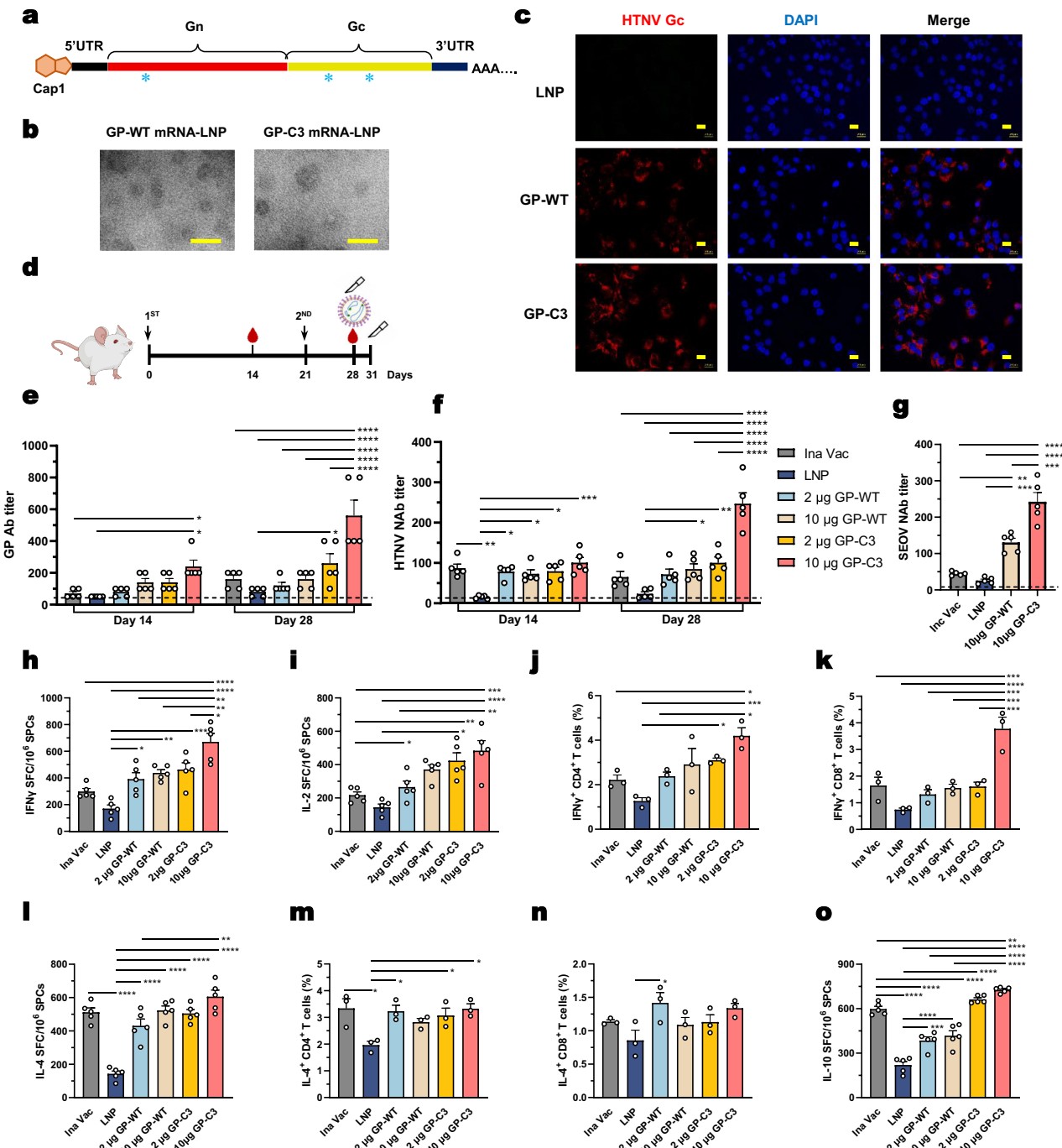

**Fig. 5 | Development and immunogenicity of a prefusion-stabilized HTNV GP mRNA-LNP vaccine. a** Schematic of the prefusion-stabilized HTNV GP mRNA construct, with domains labeled: Gn (red), Gc (yellow), UTRs (black), Cap1 structure (5′ cap). The engineered cysteines for disulfide stabilization are marked by blue asterisks. **b** Representative negative-stain transmission electron microscopy images of GP-WT (left) and GP-C3 (right) mRNA-LNP particles. Scale bar, 100 nm. **c** In vitro expression validation. Huh7 cells were treated with GP-WT or GP-C3 mRNA-LNP or empty LNP, and protein expression was detected using an HTNV Gc-specific monoclonal antibody (3G1). Scale bar, 20 μm. **d** Immunization and challenge scheme. Female BALB/c mice (6–8 weeks old) received two i.m. doses of vaccine at a 3-week interval. Blood and spleens were collected at the indicated times for immune analysis. For the challenge, mice were inoculated with HTNV on day 28 and sacrificed on day 31. **e, f** Humoral immune responses post-immunization: HTNV GP-

specific IgG titers (**e**) and HTNV-neutralizing antibody (NAb) titers (**f**) at days 14 and 28. The dashed line indicates the detection limit. **g** Cross-neutralizing antibody titers against Seoul virus (SEOV) in serum at day 28. **h–o** Cellular immune responses in splenocytes at day 28. Frequencies of IFN-γ− (**h**) and IL-2−secreting cells (**i**) (ELISpot), IFN-γ⁺ CD4⁺ (**j**) and CD8⁺ T cells (**k**), IL-4−secreting cells (**l**) (ELISpot), IL-4⁺ CD4⁺ (**m**) and CD8⁺ T cells (**n**), and IL-10−secreting cells (**o**) (ELISpot) in response to HTNV-GP peptide stimulation. Data are presented as mean ± SEM. For ELIZA and ELISpot assays (**e–i**, **l**, p), $n = 5$ mice per group. For (**b**, **c**), this experiment was independently repeated three times with similar results. For flow cytometry analyses (**j**, **k**, **m**, **n**), $n = 3$ mice per group. Statistical significance was determined by two-way ANOVA (**e**, **f**) or one-way ANOVA (**g–p**) with Tukey's multiple comparisons test. *$P < 0.05$, **$P < 0.01$, ***$P < 0.001$, ****$P < 0.0001$. Exact $P$-values are provided in the Source Data file. Source data are provided as a Source Data file.

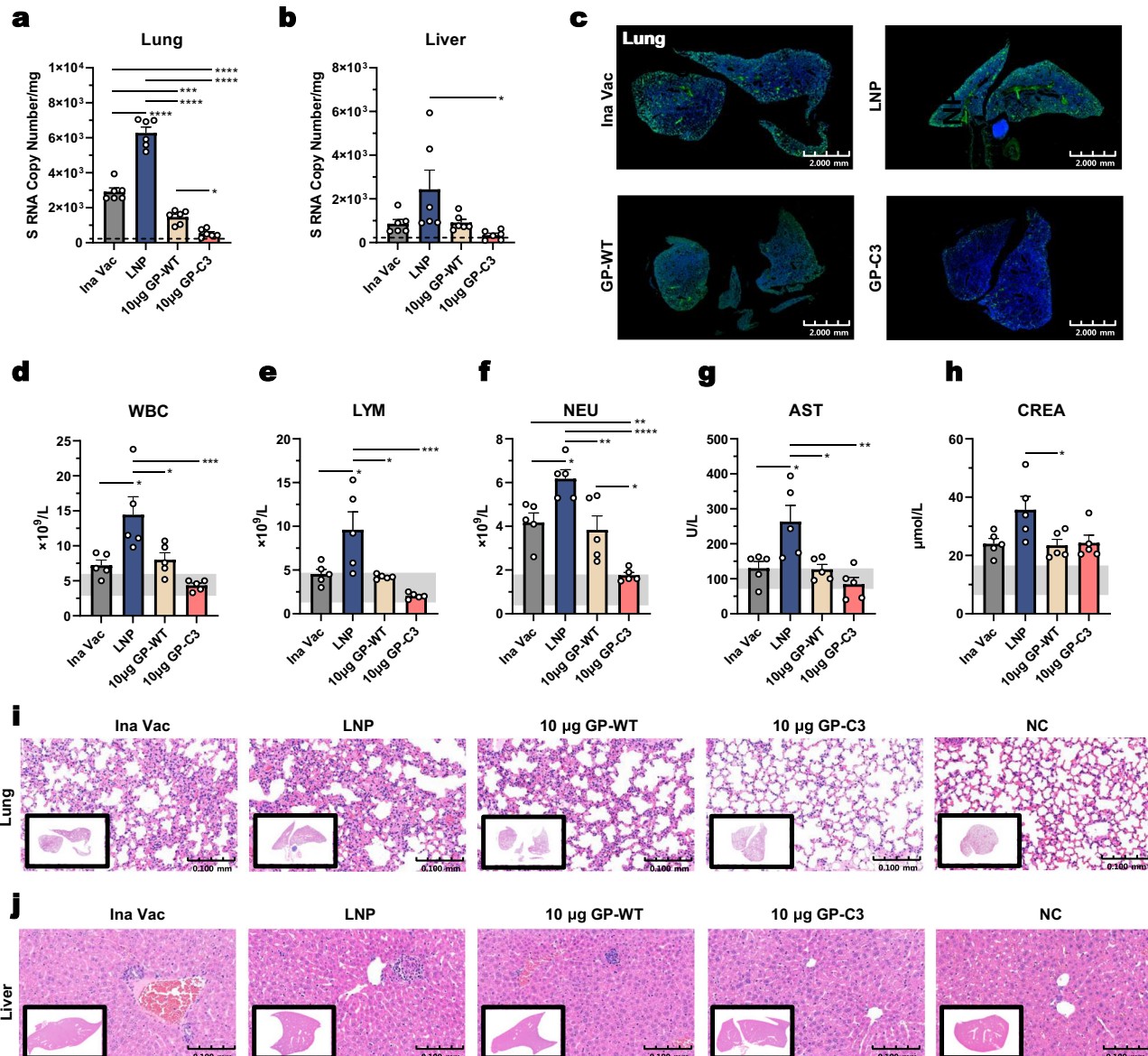

**Fig. 6 | Protective efficacy of the prefusion-stabilized HTNV GP mRNA-LNP vaccine against viral challenge. a, b** Viral RNA loads (copies/mg) in lung (**a**) and liver (**b**) tissues were quantified by qRT-PCR. The dashed line represents the detection limit. **c** Representative fluorescence immunohistochemistry images of lung sections showing HTNV nucleoprotein (NP, green) and nuclei (DAPI, blue). Scale bar, 2 mm. **d–h** Whole blood counts and serum biochemistry analyses at 3 days post-challenge; white blood cells (WBC, **d**), lymphocytes (LYM, **e**), neutrophils (NEU, f), aspartate aminotransferase (AST, **g**), and creatinine (CREA, **h**).

**i, j** Representative H&E-stained sections of lung (**i**) and liver (**j**) from infected mice, showing pathological changes. Scale bar, 100 μm. Data are presented as mean ± SEM. For viral RNA loads (**a**, **b**), $n = 6$ mice per group; for blood counts and biochemistry (**d–h**), $n = 5$ mice per group. Statistical significance was determined by one-way ANOVA with Tukey's multiple comparisons test. *$P < 0.05$, **$P < 0.01$, ***$P < 0.001$, ****$P < 0.0001$. Exact $P$-values are provided in the Source Data file. Source data are provided as a Source Data file.

As expected[20,38], a homologous booster with the inactivated vaccine increased titers of both HTNV GP-specific IgG and NAbs (Fig. 8b, c). Notably, heterologous boosting with all nucleic acid vaccines further enhanced these responses. The GP-C3 mRNA-LNP booster elicited the highest levels of both GP-specific antibodies and NAbs among all groups (Fig. 8b, c). Interestingly, the delivery platform significantly impacted the outcome, as boosting with the GP-C3-based DNA vaccine resulted in lower NAb titers than with the GP-WT mRNA vaccine, likely reflecting the adjuvant effect of the LNP system. Critically, a single GP-C3 mRNA-LNP booster elevated NAb titers to a level comparable to that achieved by a full primary course of GP-C3 mRNA vaccination (Fig. 8c).

We further evaluated the cellular immune responses. The GP-C3 mRNA-LNP booster elicited higher frequencies of IFN-γ-secreting CD4+ and CD8+ T cells compared to the inactivated vaccine booster, although the difference compared to the GP-WT mRNA or GP-C3 DNA vaccines was not statistically significant (Fig. 8d, e, Supplementary Fig. 8a, b). A similar pattern was observed for IL-4-secreting CD4+ and CD8+ T cells, with the 10 μg GP-C3 mRNA dose showing the most pronounced effect (Fig. 8f, g, Supplementary Fig. 8c, d).

Collectively, a single booster dose of the prefusion-stabilized HTNV GP mRNA-LNP vaccine potently recalls and expands immunity primed by suboptimal inactivated vaccines, achieving neutralizing antibody levels equivalent to a full primary mRNA vaccination course (Fig. 9).

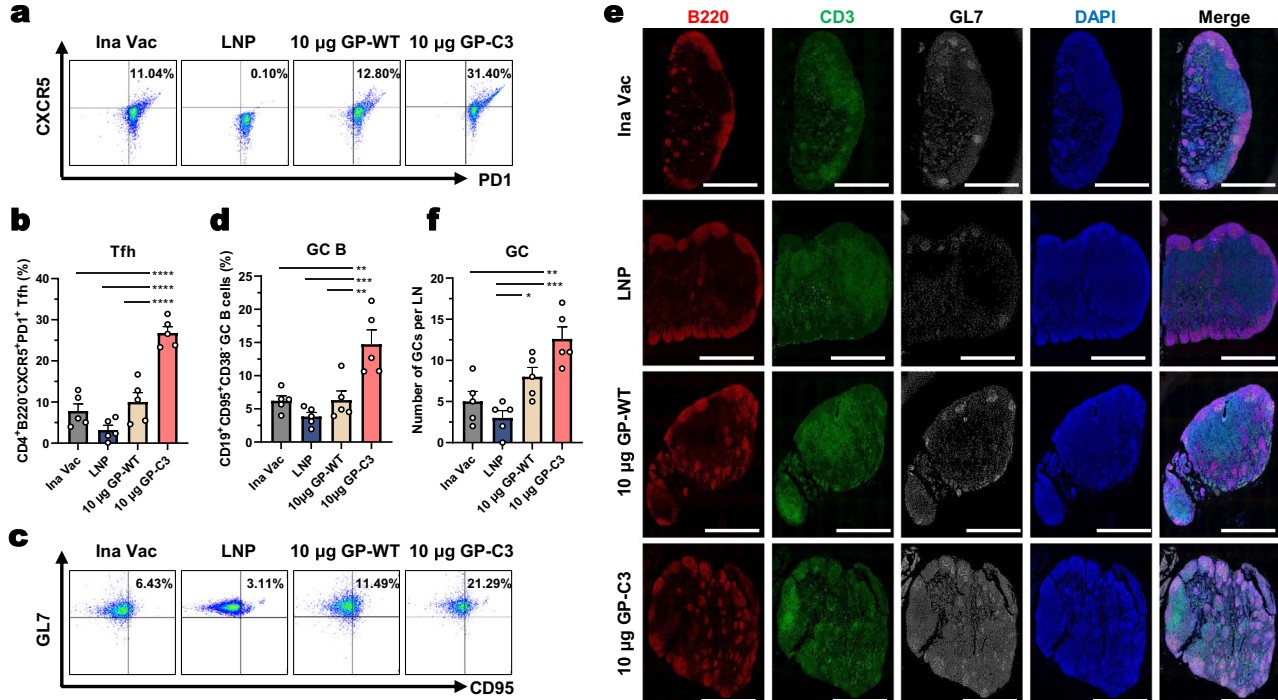

**Fig. 7 | Prefusion-stabilized HTNV GP mRNA-LNP vaccine elicits robust GC responses. a, b** Analysis of Tfh cells in draining LNs; representative flow cytometry plots (**a**) and quantification of Tfh cell frequency (**b**). **c, d** Analysis of GC B cells; representative flow cytometry plots (**c**) and quantification of GC B cell frequency (**d**). **e, f** Immunofluorescence analysis of germinal centers in lymph node sections stained for B220 (red), CD3 (green), and GL7 (white) (**e**). Scale bar, 500 μm.

Quantification of the average number of GCs per group is shown in (**f**). Data are presented as mean ± SEM, $n = 5$ mice per group. Statistical significance was determined by one-way ANOVA with Tukey's multiple comparisons test. *$P < 0.05$, **$P < 0.01$, ***$P < 0.001$, ****$P < 0.0001$. Exact $P$-values are provided in the Source Data file. Source data are provided as a Source Data file.

## Discussion

Hantaviruses remain a major global public health threat. NAbs are pivotal for protection, as demonstrated by numerous studies[25–27,39–45]. However, inactivated vaccines currently deployed in Old World hantavirus-endemic areas often fail to elicit robust and durable NAb responses[36,37]. This shortcoming, combined with environmental changes expanding risk areas[46], underscores the urgent need for next-generation vaccines capable of inducing high-titer, sustained NAbs.

A key bottleneck lies in the metastable nature of viral envelope glycoproteins. The prefusion conformation presents the most critical neutralizing epitopes, but these proteins readily shift to a post-fusion state. Vaccines presenting post-fusion antigens may elicit non-neutralizing or even potentially harmful antibodies, as tragically demonstrated by early formalin-inactivated RSV vaccines[47,48]. Rational design of prefusion-stabilized antigens has advanced next-generation vaccine development for viruses like RSV and SARS-CoV-2[29,30,49–53]. The recent elucidation of the hantavirus glycoprotein spike structure[12], provided the blueprint for applying this strategy to hantavirus, including HTNV.

In this study, we engineered disulfide bonds to stabilize the HTNV GP in its prefusion conformation. The GP-C3 construct, which incorporating two distinct stabilizing disulfide bonds between Gn-Gc and within Gc-Gc interfaces (Fig. 1a), emerged as a superior immunogen. It consistently induced the highest levels of NAbs across both DNA and mRNA-LNP platforms. This demonstrates that prefusion stabilization is a potent strategy for enhancing hantavirus vaccine immunogenicity. The superior performance of GP-C3 over the single-mutant constructs (GP-C1, GP-C2) suggests that optimal stabilization requires multi-point constraints to effectively limit conformational flexibility.

A detailed analysis of the cross-neutralizing antibody response against SEOV revealed an important nuance. While GP-C2 induced lower titers than GP-C1 or wild-type GP at day 14, its activity increased substantially by day 70. The distinct conservation patterns of the glycoprotein subunits may account for this observation. The Gc subunit, particularly the fusion loop region, is relatively conserved among orthohantaviruses, whereas Gn exhibits greater sequence diversity[54]. The GP-C2 mutation (S291C/T731C), which introduces a disulfide bond primarily between Gn and Gc, could initially perturb the presentation of key cross-reactive epitopes that depend on the conserved Gc scaffold. Over time, affinity maturation may overcome this barrier, enabling the recruitment of B cell clones targeting these conserved sites. In stark contrast, GP-C3 elicited high and sustained cross-neutralizing titers against SEOV at both early and late time points. This indicates that the GP-C3 design optimally preserves the key conformational epitopes necessary for rapid and broad cross-neutralization, solidifying its status as the ideal candidate for a broadly protective vaccine.

We first employed the DNA vaccine platform for antigen screening, a choice informed by its flexibility and the recognized challenges in recombinant production of the full-length HTNV GP spike. DNA vaccines offer benefits in stability and manufacturing scalability[55]. However, their immunogenicity, particularly for NAb induction, can be modest due to inefficient in vivo delivery and lower antigen expression[56–58]. Despite these inherent limitations, our prefusion-stabilized HTNV GP DNA vaccine induced a durable NAb response maintained without attenuation for six months, and a robust cellular immune response that evolved from an initial Th1-skewed to a sustained Th2 profile—the latter being crucial for long-lived humoral immunity. Strategies to enhance DNA vaccine efficacy, such as improved delivery devices validated in HFRS vaccine trials[59] or MHC-II targeting approaches[18,19,60], could further augment this platform.

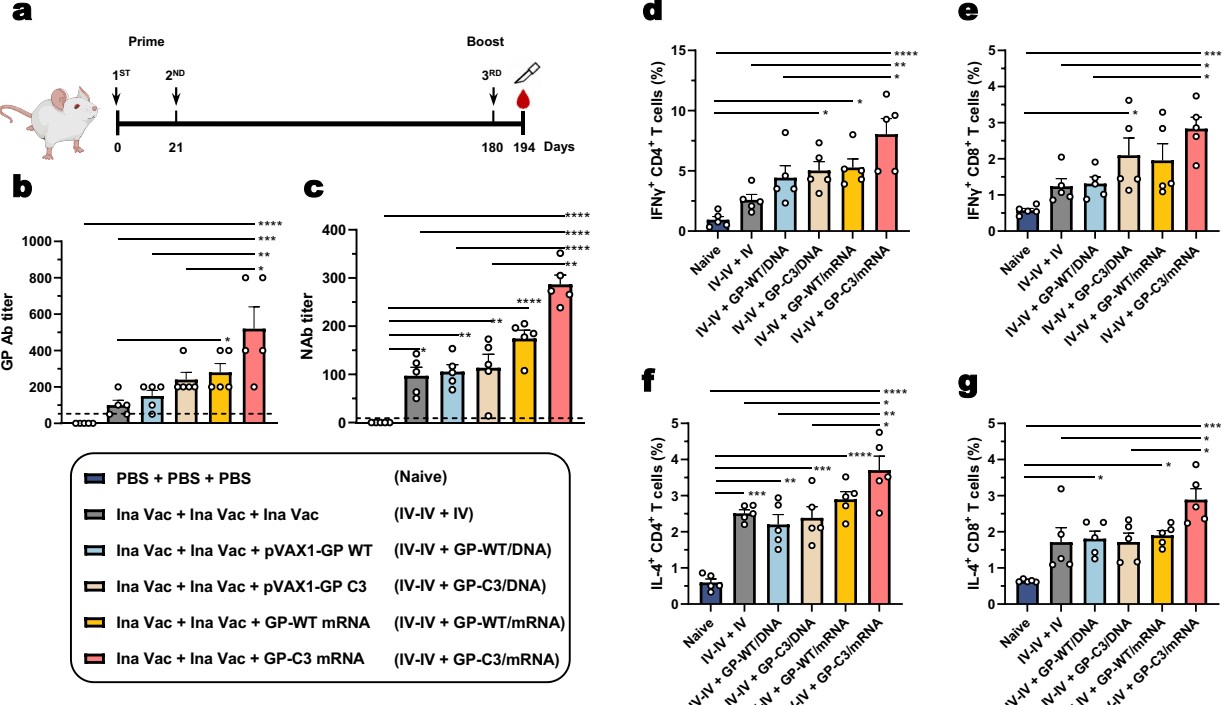

**Fig. 8 | Prefusion-stabilized HTNV GP mRNA-LNP vaccine significantly boosts immunity primed by inactivated vaccine. a** Prime-boost immunization scheme. Female BALB/c mice (6–8 weeks old) were primed with two doses of inactivated vaccine at a 3-week interval. At day 180, mice received a booster immunization with the indicated formulations. Blood and spleen were collected on day 194 for analysis. **b, c** Humoral immune responses at day 194; HTNV GP-specific IgG titers (**b**) and HTNV-neutralizing antibody (NAb) titers (**c**). The dashed line indicates the detection limit. **d–g** Cellular immune responses in splenocytes at day 194. Frequencies of IFN-γ⁺ CD4⁺ (**d**) and CD8⁺ T cells (**e**), and IL-4⁺ CD4⁺ (**f**) and CD8⁺ T cells (**g**) in response to HTNV-GP peptide stimulation. Data are presented as mean ± SEM, $n = 5$ mice per group. Statistical significance was determined by one-way ANOVA with Tukey's multiple comparisons test. *$P < 0.05$, **$P < 0.01$, ***$P < 0.001$, ****$P < 0.0001$. Exact $P$-values are provided in the Source Data file. Source data are provided as a Source Data file.

To leverage a platform with higher inherent immunogenicity, we evaluated GP-C3 in the clinically advanced mRNA-LNP format. Our study provides a direct comparison of these two nucleic acid vaccine platforms. Nucleoside-modified mRNA-LNP vaccines offer high translational efficiency and self-adjuvanting properties. The ionizable lipid component of LNPs can promote strong Tfh and GC B cell responses[61,62]. As anticipated, the GP-C3 mRNA-LNP vaccine elicited higher peak NAb titers than its DNA counterpart and triggered a more potent GC reaction, characterized by greater expansion of Tfh and GC B cells. This robust GC response, essential for antibody affinity maturation and memory formation[63], is likely facilitated by the stable, nanoparticle-like structure of the prefusion-stabilized spike, promoting its retention and display by follicular dendritic cells[64]. This outcome stands in contrast to a recent study in which delivery of the wild-type HTNV glycoprotein via alternative nucleic acid platforms elicited only modest NAb titers, underscoring the critical contribution of our prefusion-stabilization strategy to achieving potent humoral immunity[65]. The comparative analysis, summarized in Supplementary Table 1, underscores that while both platforms are effective with the GP-C3 immunogen, the mRNA-LNP formulation elicits a quantitatively stronger and more rapid humoral and GC response, highlighting the synergy between superior antigen design and an advanced delivery system.

A key translational finding is the exceptional efficacy of the GP-C3 mRNA-LNP vaccine as a heterologous booster. In a model mimicking pre-existing immunity from suboptimal inactivated vaccines, a single booster dose of GP-C3 mRNA-LNP rapidly recalled and amplified the immune response, elevating NAb titers to levels matching a full primary mRNA vaccination course. This strategy offers a promising path to enhance protection in populations already vaccinated with current inactivated vaccines.

We acknowledge certain limitations of this study. First, our evaluation relied on a murine challenge model. While rodents are natural hosts for hantaviruses, the ideal animal model for OWHs like HTNV remains lacking, unlike the Syrian hamster model available for some NWHs. Our high-dose i.m. challenge model was designed to induce acute infection for clear efficacy assessment, but it does not fully replicate natural human aerosol exposure, which may affect the translational relevance of our findings. Second, the timepoint for post-challenge analysis was chosen based on the established viral kinetics in murine models, where peak viral load in target organs typically occurs around day 3. While extended observation could provide additional clinical data, our primary endpoint was the quantification of peak viral replication, a standard metric in HTNV vaccine studies. Finally, technical hurdles in producing full-length, prefusion-stabilized HTNV GP of sufficient quality have so far precluded the generation of probes for detailed GP-specific B cell receptor analysis—a recognized challenge in the field. Nonetheless, our comprehensive functional readouts—high-titer and durable NAbs, robust GC responses, and solid protection against challenge—collectively demonstrate the success of our rational vaccine design strategy.

Future work will be essential to advance this promising candidate. First, the cross-protective breadth of GP-C3 against other clinically relevant Old World hantaviruses (e.g., DOBV, PUUV) requires empirical determination. Second, and equally critical, comprehensive safety evaluations—including specific assessment for the potential risk of antibody-dependent enhancement (ADE) upon exposure to heterologous hantaviruses—will be a mandatory component of the advanced

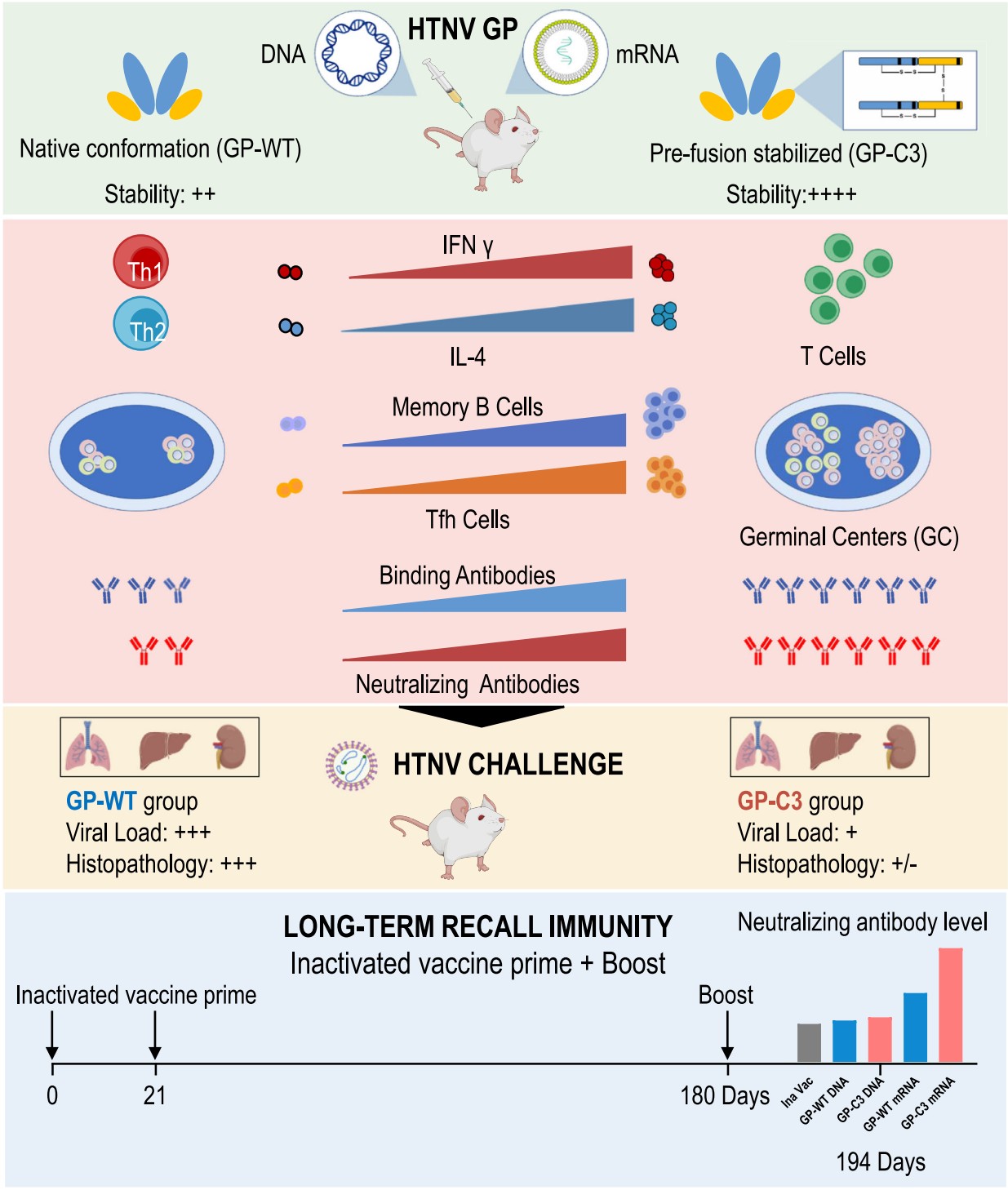

**Fig. 9 | Graphical abstract.** Prefusion-stabilized HTNV GP nucleic acid vaccines (DNA and mRNA-LNP) elicit robust humoral, cellular, and germinal center responses, conferring protection against viral challenge by reducing tissue viral loads and mitigating pathological injury. The mRNA-LNP vaccine further serves as an effective booster to amplify pre-existing immunity primed by inactivated vaccines.

preclinical and clinical development pathway[66,67]. Finally, the prefusion stabilization strategy validated here provides a blueprint for designing novel immunogens against other global threats, including New World hantaviruses such as ANDV.

In conclusion, the integration of structure-based prefusion stabilization with modern nucleic acid delivery platforms has yielded a highly promising vaccine candidate against HTNV. The GP-C3 immunogen, particularly when delivered via mRNA-LNP, elicits a potent, broad, and persistent immune response and demonstrates efficacy both as a primary vaccine and as a heterologous booster, establishing a strong foundation for the development of next-generation hantavirus vaccines.

## Methods

### Ethics statement

The experiments were performed in accordance with the recommendations of the Guide for the Care and Use of Laboratory Animals of the Ministry of Science and Technology of China. The protocols were approved by the Animal Ethics Committee of the Air Force Medical University (Fourth Military Medical University; approval no. FMMU-20200403).

### Cells, plasmids, viruses

Vero E6 (ATCC, CRL-1586), Huh7 (Procell, Wuhan, China; CL-0120), and HEK-293T (Clontech, now Takara Bio, 632180) cells were maintained in Dulbecco's modified Eagle's medium (Sigma-Aldrich, St. Louis, MO, USA) supplemented with l-glutamine, sodium pyruvate, and 10% fetal bovine serum (Sigma-Aldrich). The GP gene of HTNV strain 76–118 (GenBank: NC_005219.1) was codon-optimized and synthesized by GenScript (Nanjing, China). The gene was cloned into the pCAGGS mammalian expression vector using the NovoRec® plus One Step PCR Cloning Kit (Novoprotein, Suzhou, China) to generate a construct with a C-terminal Myc tag, following the manufacturer's instructions[68]. Internal signal peptide mutant WAASR was generated through site-directed mutagenesis using primer pairs WAASR-F and WAASR-R and amplifying the full-length pCAGGS-GP plasmid. After digesting with *Dpn* I, the prefusion-stabilized HTNV GP was obtained via cysteine mutations to generate additional disulfide bonds. Full-length cysteine-mutated GP was obtained through overlapping PCR by amplifying each mutated cysteine upstream and downstream segments. Double and triple cysteine mutations were generated based on a sequencing-verified non-mutation template. The resulting multiple pCAGGS-GP plasmids were Sanger-sequencing verified and subcloned into the pVAX1 vector. Thus, the pCAGGS vector was utilized for in vitro assays requiring high transient expression, while the pVAX1 vector, a backbone optimized for DNA vaccines, was employed for all animal immunization studies.

HTNV (strain 76–118) and SEOV (strain SR-11) stocks were propagated in Vero E6 cells. The HFRS bivalent-inactivated vaccine (HANPUWEI™) used in this study was produced by the Changchun Research Institute of Biological Products Co., Ltd. (Changchun, China). The inactivated vaccine was produced from inactivated and purified HTNV PS-6 and SEOV L-99 strains propagated in hamster kidney cells. Mice were vaccinated with 10 μg of HTNV inactivated vaccine (the manufacturer's instructions were referred to for the immunization dose), which was diluted in phosphate-buffered saline (PBS).

### mRNA synthesis, LNP encapsulation, and evaluation

mRNA was synthesized in vitro using T7 RNA polymerase-mediated transcription from a linearized DNA template derived from the plasmid pUC-simple (Sangon Biotech, Shanghai, China). The template encodes codon-optimized HTNV GP with or without a triple cysteine mutation, incorporating the 5′- and 3′-untranslated regions and a poly-A tail. Lipid nanoparticle (LNP) formulations were prepared following a standard procedure previously described for siRNAs[69]. Briefly, ethanol-dissolved ionizable lipid SM102, 1, 2-distearoyl-sn-glycero-3-phosphocholine (DSPC), cholesterol, and DMG-PEG 2000 were combined in a molar ratio of 50:10:38.5:1.5. The lipid mixture was combined with mRNA at a ratio of 1:2 in 20 mM citrate buffer (pH 4.0) using a T-mixer. Formulations were dialyzed with PBS (pH 7.4), concentrated, passed through a 0.22 μm sterile filter, and stored at −80 °C until use. All formulations were tested for particle size, distribution, and encapsulation. The resultant mRNA-LNPs were deposited on carbon-coated Formvar grids, stained with 1% phosphotungstic acid, and examined using transmission electron microscopy. Huh7 cells were seeded in 24-well plates containing coverslips ($5 \times 10^5$ cells/well). After 18 h, the cells were treated with mRNA-LNPs or empty LNPs. Six hours later, the spent

medium was replaced with Opti-MEM™ I reduced-serum medium (Thermo Fisher Scientific, Waltham, MA, USA). Twenty-four hours later, the cells were fixed with 4% paraformaldehyde (PFA), permeabilized with 0.5% Triton X-100, blocked with 3% bovine serum albumin (BSA), and incubated with the HTNV Gc-specific monoclonal antibody 3G1, and then with Cy3-conjugated goat anti-mouse IgG. Finally, the coverslips were observed using a BX60 fluorescence microscope (Olympus).

### Native-PAGE, SDS-PAGE, and Western blotting

Plasmids bearing each HTNV GP construct were transfected into HEK-293T cells using the Hieff Trans Liposomal Transfection Reagent (Yeasen Biotech, Shanghai, China) according to the manufacturer's instructions. After 24 h, the cells were harvested and lysed on ice with a radioimmunoprecipitation assay lysis buffer (Beyotime Biotechnology, Shanghai, China) for 30 min. The lysate was clarified through centrifugation at $12,000 \times g$ for 5 min. The protein levels in the supernatant were quantified using a bicinchoninic acid assay kit (Thermo Fisher Scientific, Waltham, MA, USA), and equal amounts of protein in each group were incubated with anti-c-Myc magnetic beads (MedChemExpress, Monmouth Junction, NJ, USA) overnight. After washing thrice with IP lysis buffer, each sample was separated, and one copy of each sample was directly loaded onto a 3–20% gradient polyacrylamide gel (GenScript) and subjected to electrophoresis under non-denaturing, non-reducing conditions (Native-PAGE). The other copy of each sample, added with 5% (v/v) β-Mercaptoethanol (β-ME) and boiled, was separated using SDS-PAGE. Both proteins were transferred to polyvinylidene fluoride membranes and blocked in tris-buffered saline containing 5% nonfat powdered milk. Membranes were blotted with rabbit antibodies against Myc-tag (Sangon Biotech, Shanghai, China) at 1:5000 dilution and horseradish peroxidase (HRP)-conjugated goat anti-rabbit IgG (Sangon Biotech) at 1:10,000 dilution. After washing with PBS twice, the membranes were incubated with high-sensitivity luminol (Abbkine Biotechnology, Atlanta, GA, USA) and imaged using an electrochemiluminescence system (Tanon, Shanghai, China).

### Syncytia formation assay

WT and prefusion constructs of HTNV GP were subcloned into the high expression pCAGGS vector by truncating the C-terminal six amino acids (ΔC6), and each plasmid was co-transfected with pCAGGS-ITGB3 and pCAGGS-GFP at a ratio of 1:1:1 into HEK-293T cells, with a total of 1 μg per well in a 24-well plate. After 24 h, the culture medium was replaced with a citric acid solution (pH 5.0) and incubated at 37 °C for 30 min. Subsequently, the number of syncytia formed through cell fusion was visualized using an inverted fluorescence microscope (Olympus IX71, Tokyo, Japan). The degree of membrane fusion for the observation field in each group was calculated by averaging the number of multinucleated cells in three independent biological replicates of that group.

### DNA vaccine immunization regime

All vaccine immunogenicity-related experiments were performed using 6–8-week-old female BALB/c mice purchased from the Animal Center of Air Force Medical University. Which represent a standard and well-characterized model for the initial proof-of-concept evaluation of vaccine-induced immune responses. The mice were maintained in individually ventilated cages under regular ambient room temperature (21 °C; RT), 60% humidity, and a 12 h/12 h light/dark cycle, and were randomly divided into seven groups (10–12 mice per group): six DNA-immunized groups (Vector, WT, WAASR, GP-C1, GP-C2, and GP-C3) and an HFRS-inactivated vaccine control group. Mice were immunized with 50 μg (25 μg per leg) of the DNA vaccine through i.m. injection on days 0, 28, and 56. Serum was collected 2 weeks after each immunization and stored at −80 °C until specific antibody IgG and NAb assays were

performed to assess the level of vaccine-induced humoral immune response. Two weeks after the third immunization, five mice from each group were randomly selected, and splenic lymphocytes were isolated for ELISpot and intracellular cytokine staining to evaluate the level of vaccine-induced T cell immune response. The remaining five or six mice in each group were subjected to HTNV strain 76–118 challenge with $5 \times 10^5$ FFUs via i.m. injection. Three days post-infection—corresponding to the peak of viral replication in the murine model—tissues from various organs were collected and examined for pathological damage and viral load. For the long-term immunization regime, mice were immunization as above, while the serum and splenocytes were collected at day 180.

## mRNA-LNP vaccine immunization regime

For the short-term immunization regime, groups of 6-to-8-week-old female BALB/c mice were i.m. immunized with WT or GP-C3 mRNA-LNP (2 μg or 10 μg, $n = 10$), or LNP placebo (10 μg, $n = 10$) in 100 μL PBS using an insulin syringe (BD Biosciences, San Jose, CA, USA). An HFRS-inactivated vaccine was used as a positive control. Each mouse received a booster of an equal amount and type of inactivated vaccine or mRNA-LNP on day 21 post-initial immunization. Serum samples were collected on days 14 and 28 post-immunization to detect HTNV GP-specific IgG and NAb responses. At day 28 post-initial immunization, the spleen tissues were collected to evaluate cellular immune responses via ELISpot and flow cytometry. HTNV challenge and tissues, as well as sera collection, as it does in the DNA vaccine regime. For the prime-boost immunization regime, mice were i.m. immunized with two doses of the HFRS-inactivated vaccine at a three-week interval and boosted with an equal dose of inactivated vaccine, 50 μg of GP-WT or GP-C3 DNA vaccine, or 10 μg of GP-WT or GP-C3 mRNA-LNP on day 180 post-initial immunization. At day 194 post-initial immunization, serum and spleen tissues were collected to evaluate HTNV GP-specific IgG and NAb responses as well as cellular immune responses.

## Enzyme-linked immunosorbent assay (ELIZA) based HTNV GP-specific antibody titer evaluation

Lab-defined ELIZA was used to determine HTNV GP-specific antibody titers. A previously rescued recombinant vesicular stomatitis virus lacking VSV-G but expressing HTNV GP (rVSV-HTNV GP) was used as the coating antigen[70]. Briefly, 96-well ELIZA plates were coated with 2.0 μg of iodixanol-purified rVSV-HTNV-GP in 100 μL ELIZA Coating Buffer per well overnight at 4 °C. The plates were then washed with PBS and blocked with 3% bovine serum albumin in PBS for 1 h at RT. Mouse serum was 2-fold serially diluted starting at a 1:50 dilution in blocking buffer, added to the washed ELIZA plates (100 μL/well), and incubated for 2 h at 37 °C. Following three washes with PBS, the plates were incubated with horseradish peroxidase (HRP)-conjugated goat anti-mouse IgG (1:5,000; Sangon) for 60 min at 37 °C. After washing, the plates were incubated with 3,3′,5,5′-tetramethylbenzidine substrate solution (TIANGEN, Beijing, China) for 15 min at 37 °C after five PBS washes, followed by quenching with 2 M sulfuric acid. The absorbance was read at 450 nm using a microplate reader (Bio-Tek, Winooski, VT, USA). Results with a positive/negative ratio >2.1 were considered positive[71]. Antibody titers were defined as the reciprocal of the maximum serum dilution with a positive response.

## Focus reduction neutralization test (FRNT)

The focus reduction neutralization test (FRNT) was performed based on a previously described protocol[72], with the following modifications: Vero E6 cells were seeded in 96-well plates ($1 \times 10^4$ cells/well) and incubated for approximately 18 h until reaching over 90% confluence. Mouse serum was serially diluted two-fold in PBS, starting at a ratio of 1:10. The diluted sera were then mixed with 100 FFUs of HTNV (strain 76–118) or SEOV (strain SR-11) at a 1:1 (v/v) ratio in 100 μL Dulbecco's modified Eagle's medium (DMEM) containing 2% fetal bovine serum

(FBS), and incubated at 37 °C for 1 h. After removing the mixture, the cells in the plates were overlaid with DMEM containing 2% FBS and 1.6% (w/v) carboxymethylcellulose sodium salt. After further incubation at 37 °C for 7 days, the cells were fixed with 4% PFA, permeabilized with 0.5% Triton X-100, and incubated with an HRP-conjugated HTNV NP-specific 1A8 monoclonal antibody for approximately 18 h at 4 °C. Foci representing Virus-infected cells were quantified using an AID-ELISPOT reader (Straßberg, Germany) after visualization with the precipitated TMB substrate. The neutralizing antibody titer was defined as the maximum serum dilution that conferred a 50% reduction in the formed foci ($FRNT_{50}$) and calculated using the Spearman–Karber method[73].

## Enzyme-linked immunospot (ELISpot) assay

Cellular immune responses, characterized by the number of IFN-γ-, IL-2-, IL-4-, and IL-10-secreting splenocytes in vaccinated mice, were assessed using pre-coated ELISPOT kits (MabTech, NackaStrand, Sweden) following the manufacturer's protocols. Briefly, splenocytes were collected from the immunized mice, and red blood cells were removed. Freshly isolated splenocytes were resuspended in 10% RPMI-1640 medium and transferred to the coated ELISpot plates at a density of $1 \times 10^6$ cells/well. The cells were then stimulated with a 10 μg/mL HTNV Gn/Gc peptide pool (Supplementary Tables 2, 3) for 24 h at 37 °C. Positive control stimulation was performed using 1 μg/mL concanavalin A (ConA, Sigma), while negative control stimulation was conducted using RPMI-1640 medium alone. After 24 h of incubation at 37 °C, 5% $CO_2$, biotinylated anti-mouse IFN-γ, IL-2, IL-4, or IL-10 antibodies were added to each well and incubated for 2 h at RT.

Streptavidin-horseradish peroxidase was then added and incubated for 1 h. An AEC substrate solution was subsequently added, and the reaction was stopped by rinsing the plates with deionized water. Air-dried plates were analyzed using an AID-ELISPOT reader, and the number of spot-forming cells (SFCs) per $1 \times 10^6$ cells was calculated.

## T-cell intracellular cytokine staining

To assess T lymphocyte proliferation in immunized mice, $1 \times 10^6$ splenocytes were transferred into each well of a 48-well plate and stimulated with a 10 μg/mL HTNV Gn/Gc peptide pool (Tables S1, S2) for 2 h at 37 °C. Subsequently, 10 μg/mL brefeldin A was added to each well and incubated for 4 h. Negative and positive controls were included, using dimethyl sulfoxide and a cell activation cocktail (with brefeldin A), respectively. Following two washes with PBS, splenocytes were stained with Zombie Aqua™ Fixable Viability (BioLegend, San Diego, CA, USA) and fluorescent-conjugated antibodies against the surface markers CD3 (PE/Cyanine7; BioLegend), CD4 (FITC; BioLegend), and CD8 (APC/Fire 750™; BioLegend). The splenocytes were then fixed with a permeabilization buffer (Thermo Fisher) and stained for IFN-γ (APC; eBioscience, Santiago, CA, USA), TNFα (PE, eBioscience) or IL-4 (PerCP-eFluor 710; eBioscience). All labeled lymphocytes were analyzed using a CytoFLEX flow cytometer (Beckman Colter, Pasadena, CA, USA) and CytoExpert software (Beckman Colter).

## Measurement of viral burden in challenged mouse tissues

Mouse tissues in the RNA preservation solution were weighed and transferred to a new tube, to which TRIzol Universal RNA extraction solution (TIANGEN) was added. The tissues were then homogenized with sterile stainless-steel beads using Tissue Lyser II (QIAGEN, Dusseldorf, Germany), and RNA was isolated according to the manufacturer's instructions. One microgram of total RNA was reverse-transcribed into cDNA and amplified using SYBR Green Master Mix (Yeasen Biotech). Viral RNA levels were determined through qRT-PCR using HTNV-S-specific primer pairs (forward: 5′-TCTAGTTGTATCCC-CATCGACTG-3′; reverse: 5′-ACATGCGGAATACAATTATGGC-3′) and normalized to tissue weight and a standard curve[74].

## Tissue histological immunofluorescence and histopathology

Three days post-HTNV challenge, whole blood samples from each mouse were collected. The livers, spleens, lungs, and kidneys of the mice were excised and weighed. Part of the tissue was preserved in a non-freezing tissue RNA preservation solution, and the remaining tissues were cut into small pieces and placed in 4% PFA for hematoxylin and eosin staining and tissue immunofluorescence. Tissues fixed in 4% PFA for 24 h were processed into paraffin-embedded microtomy sections. For fluorescence immunohistochemistry, paraffin sections were sequentially incubated with the mAb-1A8 antibody, and Cy3-conjugated goat anti-mouse IgG (DNA vaccine regime) or FITC-conjugated goat anti-mouse IgG (mRNA-LNP regime), and the nuclei were stained with DAPI. Finally, the slices were sealed with an anti-fluorescence quenching sealing agent (Solarbio, Beijing, China). Images were captured using a Pannoramic MIDI instrument (Thermo Fisher) and analyzed using CaseViewer V2.4.0 software (3D HISTECH, Budapest, Hungary). For histopathological detection, paraffin-embedded microtome sections from mouse lung, liver, kidney, and spleen tissues were deparaffinized, rehydrated, and stained with hematoxylin and eosin, followed by treatment with ethanol and xylene.

## Measurement of biochemical index

Blood from vaccinated mice was collected 3 days post-HTNV challenge and mixed with 1.5 mg/mL $K_2$-ethylenediaminetetraacetic acid (EDTA). Analysis of WBC, lymphocyte, and neutrophil counts was performed using a BC-2800vet automatic blood cell analyzer (Mindray Bio-Medical Electronics Co., Ltd., Shenzhen, China). Serum samples were analyzed to determine the levels of ALT, AST, ALB, BUN, and CREA using an automated biochemical analyzer (ChemRay 800; Rayto Life and Analytical Sciences Co., Ltd., Shenzhen, China).

## Detection of Tfh and GC B cells and GCs in situ

To detect Tfh and GC B cells, groups of 6-to-8-week-old female BALB/c mice were intramuscularly immunized with only one dose. Inguinal LNs were collected at 7 days post-immunization, and single-cell suspensions were obtained. For Tfh analysis, lymph node cells were incubated with fluorescently conjugated antibodies against CD4 (FITC; BioLegend), B220/CD45R (APC/Cy7; BioLegend), PD1 (PerCP-eFluor 710; eBioscience), and CXCR5 (PE/Cy7; BioLegend). GC B cells were labeled with CD19 (FITC; BioLegend) and CD95 (PerCP-Cy5.5). For the in situ detection of GCs, mouse inguinal LNs were harvested and fixed in 4% PFA for 30 min at RT. Snap-freezing was performed in a Tissue-Tek Cryomold using an optimal cutting temperature compound (Yeasen Biotech) in a dry ice-cooled bath of 2-methylbutane. Subsequently, 8-μm-thick tissue sections were blocked with PBS containing 5% goat serum and FcR blocking reagent (Biolegend). The sections were then stained with antibodies against CD3, B220, and GL7 for 18 h at 4 °C, and detected using TSA-dendron fluorophores. Nuclei were stained with DAPI. Images were acquired using a Zeiss LSM710 confocal microscope (Oberkochen, Badenwalburg, Germany) and processed and analyzed using CaseViewer V2.4.0 software.

## Quantification and statistical analysis

Statistical analyses were performed using the GraphPad Prism 8.0 software. Tests, number of animals (n), median values, and statistical comparison groups are shown in the figure legends. Two-way analysis of variance (ANOVA) or One-way ANOVA followed by Tukey's multiple comparisons test was performed. Statistical significance was set to $P < 0.05$ and indicated as follows: $*P < 0.05$, $**P < 0.01$, $***P < 0.001$, $****P < 0.0001$, ns: not significant.

## Reporting summary

Further information on research design is available in the Nature Portfolio Reporting Summary linked to this article.

## Data availability

All data generated in this study are provided in the Source Data file associated with this paper. No sequencing or other omics data requiring deposition in public repositories were generated. Source data are provided with this paper. All reagents will be made available upon request after the completion of a material transfer agreement. Requests can be directed to the corresponding author. Source data are provided with this paper.

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

## Acknowledgments

We thank the Center for Medical Innovation of Air Force Medical University for the technical support of the flow cytometry examination. F.L.Z. discloses support for the research of this work from the National Key Research and Development Program of China (No. 2022YFC2604200) and Key Research and Development Project of Shaanxi Province (No. 2019ZDLSF02-04). Z.K.X. discloses support for the research of this work from the National Natural Science Foundation of China (No. 82072268). W.Y. discloses support for the research of this work from the Air Force Medical University (No. 2021JSTS10). L.Z. discloses support for the research of this work from the Air Force Medical University (No. 2025KXKT115). Y.W. discloses support for the research of this work from the Air Force Medical University (No. 2022ZZXM044).

## Author contributions

Conceptualization: W.Y., Y.M.D., and F.L.Z. Methodology: W.Y., Y.M.D., Y.W., Q.Q.Y., H.Z., C.T.Y., J.W., L.Z., L.F.C., H.W.M., and H.L. Resources: C.T.Y., J.W.P., X.M.P., D.S.J., X.J.Y., X.L.J., and Y.C.D. Investigation: W.Y., Y.M.D., Q.Q.Y., H.Z., Y.W., J.W., and C.T.Y. Visualization: W.Y., Q.Q.Y., Z.K.X., Y.F.L., and F.L.Z. Funding acquisition: WY, YW, ZKX, and FLZ. Project administration: Z.K.X. and F.L.Z. Supervision: W.Y., Z.K.X., Y.F.L., and F.L.Z. Writing – original draft: W.Y., Y.M.D., and Q.Q.Y. Writing – review & editing: W.Y., Z.K.X., Y.F.L., and F.L.Z.

## Competing interests

Authors W.Y., F.L.Z., Y.D., L.F.C., H.Z., L.Z., H.L., and Z.K.X. are inventors on a provisional patent application filed in China (Application No. CN202310156260.7) related to the prefusion-stabilized HTNV glyco-protein vaccine design described in this study. All other authors declare no competing interests.
