## [Peer Review file · Nature Communications]

Prefusion-stabilized Hantaan Virus Glycoprotein Nucleic Acid Vaccine Elicits Potent Neutralizing Antibody Responses via Germinal Center Activation

Corresponding Author: Dr Wei Ye

Version 0:

Reviewer comments:

Reviewer #1

(Remarks to the Author)

HFRS caused by Old World hantaviruses leads to severe disease in humans. The inactivated vaccine currently used in endemic areas has limited efficacy, and the development of a new vaccine platform is desired. The authors introduced one to three cysteine mutations in the GP of the Hantaan virus to stabilise it in the pre-fusion state, so that the Gn and Gc of the same molecule, and /or the Gc-Gc of two molecules, are S-S linked. Mice were immunised with the DNA plasmids and/or the mRNA-encapsulated LNPs expressing these GP mutants. Mutant with S-S binding between Gn and Gc of the same molecule and between Gc-Gc of two molecules (called GP-C3) induced the strongest neutralising antibodies in all immunisation experiments. HTNV challenge experiments were also conducted in post-immunisation mice, showing that the viral load in various organs was reduced and blood markers tested were close to normal. In addition, as there are many inactivated vaccine recipients in the actual endemic area, this vaccine candidates were administered after immunisation with inactivated vaccine, and strong neutralising antibody production was observed. Furthermore, the study also showed that the lymph nodes of the immunised mice had more germinal centre B cells than others, supporting the immunological superiority of this vaccine platform.

This vaccine design appears to be very useful in the prevention of HTNV infection, but there are several points that need to be addressed. It also has a number of typos, which should be checked carefully again.

Major comments.

1. The histopathology is shown, but it is difficult to determine the degree of inflammation without uninfected individual controls. Uninfected histopathology should also be presented. Were the lungs inflated at necropsy? Also, please provide these pathology images at lowest magnification, as a supporting file. This is to deny that the images presented are the intended extraction of the best parts by the authors.
2. More details on the design and structure of the mRNA vaccine are needed, including a clear description of the Cap structure and other RNA modifications.
3. The WB image in Fig 1C shows the position of (Gn-Gc)x4, but the separation in this region is not good. Doublets are also observed. Please show (Gn-Gc)x4 using a gel with good separation in this region.
4. Transfection efficiency is not considered in the syncytial formation experiments. The vertical axis in Fig 1d uses field as the denominator, which could varies with transfection efficiency. Therefore, I believe that for a more accurate assessment, it is necessary to calibrate with transfection efficiency.
5. In the graph showing the number of viral copies in infected individuals, the vertical axis is linear, so there appears to be a very large difference between GP-C3 and other immuns, but what is actually a fraction of that when compared to vectors?
6. The authors do not state whether the mice showed any weight changes after each vaccination, or whether they developed any other symptoms that might be considered side effects. This point should be included.

Minor comments.

Line 118. 「..., while the monomer Gc level in the GP-2 and GP-C3...」, is this a mistake for GP-C1 and GP-C2? Please

verify.

2. Line 179. It would be better for the reader if the main text describes how many days after vaccination the HTNV was challenged.
3. Line 196. Fig 1f seems to be incorrect, is it correctly Fig 2? Please verify.
4. Line 202. Fig 2c seems to be incorrect, is it correctly Fig S3g? Please confirm.
5. Line 205. BUN, Fig S3i? ALB, Fig S3h?
6. Line 606. mRNA-LNP seems to be incorrect, is it correctly mRNA-LNP?
7. The GL7 in Fig 6e is too faint and no difference can be seen in print.
8. Normal values for blood test (cell counts, liver enzymes and renal function) are listed in Fig S3e, but it would be easier to read if each bar graph was shaded or lined to indicate normal values.

Reviewer #2

(Remarks to the Author)

The manuscript by Ye and coworkers reports the findings on development of a rationally designed orthohantavirus vaccine prototype. The rationale behind the study is that the vaccine currently used in the HTNV endemic regions, Hantavax, does not necessarily provide a long-lasting strong neutralizing antibody response. Hantavax, to my knowledge, is formalin-inactivated virus (the authors might perhaps include some details of the inactivation method for the vaccine?), which might lead to masking of epitopes essential for neutralization. To tackle the potential instability issues of the spike complex, the authors introduced cysteine residues to selected (based on the 3D structures of the spike complex) regions in the Gn and Gc glycoproteins. The introduction of two disulphide bridges indeed led to stabilization of the heterotetrameric HTNV spike complex based on SDS-PAGE migration, whereas introduction of a single bridge did not act as efficiently. The authors then went on to test the immunogenicity of the engineered prefusion-stabilized spike in mice using both DNA and mRNA transfection. The extensive studies performed provided evidence supporting the authors' initial hypothesis that prefusion-stabilized spike complex induced more robust neutralizing antibody response than the response resulting from immunization using inactivated virus. In general, the results seem convincing.

Some specific comments/suggestions to improve the manuscript:

- A round of language polishing might be beneficial
- Figure 1a, would it be possible to include similar schematic figures for all constructs tested (at least GP-C1, GP-C2, GP-C3)? This might help the reader to more easily appreciate the differences?
- Have the authors considered of presenting the DNA and mRNA immunization data in the same graphs? This might allow the authors to shorten the manuscript and might allow comparison between the plasmid and mRNA immunizations.

In summary, the authors present interesting data which in my opinion are based on a solid hypothesis (approach utilized in other vaccines, as the authors point out). The conclusions and claims presented appear to be supported by the experimental data. The methods fit well for the work conducted and they have been described in adequate detail.

Reviewer #3

(Remarks to the Author)

Ye et al. describe the development and testing of prefusion-stabilized Hantaan virus (HTNV) vaccines. The study involved the rational design of DNA and mRNA-LNP vaccines, as well as the evaluation of their immunogenicity and protective efficacy in mice. The vaccine formulations were analyzed for their biochemical and histopathological effects, as well as their ability to induce humoral and cellular immune responses. The study also included the measurement of viral burden in challenged mouse tissues and the assessment of serological parameters. The results demonstrated that the prefusion-stabilized HTNV GP-based nucleic acid vaccines (especially the GP-C3 one) elicited robust humoral and cellular immunity, providing potential protection against HTNV-induced histopathological injury and other pathological effects. The authors did an extensive experimental work on selecting the best vaccine regarding the one they tested, however important majors and minors points need to be addressed before considering publishing this study.

Major points

The authors have already contributed significantly to the field with their previous publications on the development of an HTNV VSV-based vaccine (npj vaccines PMID: 38341504) and Hantaan virus pathogenicity (Nature Comm PMID: 38200007). Additionally, another preprint article coming from the same University and investigating glycoprotein-based nucleic acid vaccination for the effective control of HTNV epidemics is under review at npj vaccines (<https://doi.org/10.21203/rs.3.rs-3933421/v1>). However, it's worth noting that the present study shares considerable similarity with these 2024 HTNV vaccine studies, potentially impacting the novelty of the current work.

One important limitation of the study, as acknowledged by the authors themselves, is the significant lack of evaluation of GP-specific B cells. The stability of GP-C3 and GP-C2 proteins allows for their characterization by SDS-PAGE +/- -mercaptoethanol. However, employing affinity chromatography coupled with fPLC purification could potentially enabling the recovery of GP-C3 and GP-C2 proteins, for further linkage to fluorophores, enabling specific B-cell sorting. Another possibility would be using Gn/Gc construct to make flow cytometry probes to sort GP-specific B cells. Evaluating affinity maturation of sorted B-cells could offer valuable information regarding the immunological response, including VH antibody classes, sequences, and mutations needed for effective neutralization of the virus and protection against HTNV infection.

Regarding this study, while the authors have evaluated GC B cells numbers in LNs, information about the presence of GCs at d70 or d180 post-immunization (instead of only after 7 days) and the role of GP-C3 DNA immunogen in inducing a significant humoral memory response for long-term protection against HTNV is lacking.

Furthermore, the paper evaluate both DNA vaccine and RNA LNP vaccine studies, which are presented in a manner that appears as two separate studies within a single article. Restructuring the presentation to focus on comparing GP-C3 as DNA or RNA vaccines would enhance the coherence of the narrative and aid in identifying the most effective HTNV GP vaccine candidate.

Minor points:

- Legends for Fig 1c (electrophoresis) are different from the text (line 112,113);
- Why using two different plasmids for HTNV transfection?
- Line 118, GP-C2 and GP-C1 (instead of GP-C3);
- Line 121, "Fig S1b" is maybe Fig S1c?
- Line 124, "Fig S1c" is the electrophoresis figure, not the syncytia. Maybe Fig S1b?
- For the neutralization assay of SEOV virus, why did you choose to only take d14 sera and not d70? How the SEOV Nab levels induced by stabilized GP-C1 and GP-C2 are lesser than those induced by the attenuated Ina Vac (Fig S1d)?
- Why scarifying mice after just 3 days of challenge? We generally wait for 5 days to do a post-challenge assess virus dissemination (PMID: 32667279). It would also be better to follow the wight loss for 14 days to see any sign of infection and/or protection.
- In Figure 2 and figure S3, a statistical fluorescence evaluation is need as the GP-C3 seems to be significantly lowering the viral load only in the kidney, compared to attenuated Ina Vac and WT.
- Line 196, why "Fig 1f"?
- Statistical analysis in all graphs are confusing. To make it more powerful, it would be better to have results interpretation comparing the DNA immunogens to the attenuated Ina Vac or WT, and not to the vector.
- A healthy cotrol (non-infected) for all tissue histology is missing in the comparison.
- Line 234, figure "1h" is showing IFN SFC, you mean "Fig 1g"
- Line 292, is "4d"
- Line 295, this sentence is not true as it is now well known that one of the weakest points of mRNA vaccines, especially SARS-CoV2 mRNA vaccines is that they lack on effective long term T-cell activation.
- One of the interesting parts of this study is the mixed vaccine protocol by priming with the inactivate Ina Vac and boosting with different GP construct. This part, in my opinion, is the most innovative and should've received more focus for the immune response elicited and to suggest a better way to target the immune system to a more specific and protecting HTNV immune response.

Overall, while the study presents intriguing findings, addressing the aforementioned concerns and enhancing experimental clarity and coherence is significantly needed and would substantially improve the manuscript to be considered for a publication in Nature Communications.

Version 1:

Reviewer comments:

Reviewer #1

(Remarks to the Author)

The authors have responded very carefully to the reviewers' comments, and the revised manuscript has become a very valuable. I look forward to seeing this vaccine advance toward practical application. In particular, the GP-C3 mRNA vaccine achieved an FRNT50 of approximately 1:250, suggesting that its protective efficacy is likely to be sufficient in vivo.

In this study, post-immunization sera neutralized SEOV; however, it will be important in future work to determine whether cross-neutralization extends to other Old World hantavirus species. In addition, it will be essential to demonstrate that

vaccination does not lead to antibody-dependent enhancement upon subsequent infection with heterologous hantaviruses.

Reviewer #3

(Remarks to the Author)

Thanks for the revised manuscript and the detailed replies. The part I cared most about was the potential overlap with your previous HTNV vaccine work and the other closely related paper, and I think you handled that well. Your explanation makes it clear why this is a different story and not just a repeat, especially because the core advance here is the structure guided prefusion stabilization approach and the GP C3 design, and you also make a clear distinction between a true mRNA LNP platform and other delivery setups. What helped is that you also reflected this in the manuscript text, so readers will understand the novelty without having to rely on the rebuttal.

The present revision reads cleaner and the flow is easier to follow. It is now straightforward to see how the DNA experiments were used to identify the lead construct and then how the same lead was tested in the mRNA LNP format. I also appreciate that you acknowledged the B cell level limitation and framed it as a real technical bottleneck rather than ignoring it. At this point, I feel my major concerns are addressed and I do not have additional substantive requests.

Point-by-point response to the reviewers' comments

We thank the reviewers for their time and constructive comments, which have significantly improved our manuscript. Below, we provide a point-by-point response to all comments. All changes in the manuscript have been highlighted in yellow for ease of review.

REVIEWER COMMENTS

Reviewer #1 (Remarks to the Author):

HFRS caused by Old World hantaviruses leads to severe disease in humans. The inactivated vaccine currently used in endemic areas has limited efficacy, and the development of a new vaccine platform is desired. The authors introduced one to three cysteine mutations in the GP of the Hantaan virus to stabilise it in the pre-fusion state, so that the Gn and Gc of the same molecule, and /or the Gc-Gc of two molecules, are S-S linked. Mice were immunised with the DNA plasmids and/or the mRNA-encapsulated LNPs expressing these GP mutants. Mutant with S-S binding between Gn and Gc of the same molecule and between Gc-Gc of two molecules (called GP-C3) induced the strongest neutralising antibodies in all immunisation experiments. HTNV challenge experiments were also conducted in post-immunisation mice, showing that the viral load in various organs was reduced and blood markers tested were close to normal. In addition, as there are many inactivated vaccine recipients in the actual endemic area, this vaccine candidates were administered after immunisation with inactivated vaccine, and strong neutralising antibody production was observed. Furthermore, the study also showed that the lymph nodes of the immunised mice had more germinal centre B cells than others, supporting the immunological superiority of this vaccine platform.

This vaccine design appears to be very useful in the prevention of HTNV infection, but there are several points that need to be addressed. It also has a number of typos, which should be checked carefully again.

Major comments.

1. The histopathology is shown, but it is difficult to determine the degree of inflammation without uninfected individual controls. Uninfected histopathology should also be presented. Were the lungs inflated at necropsy? Also, please provide these pathology images at lowest magnification, as a supporting file. This is to deny that the images presented are the intended extraction of the best parts by the authors.
2. More details on the design and structure of the mRNA vaccine are needed, including a clear description of the Cap structure and other RNA modifications.

3. The WB image in Fig 1C shows the position of (Gn-Gc)x4, but the separation in this region is not good. Doublets are also observed. Please show (Gn-Gc)x4 using a gel with good separation in this region.
4. Transfection efficiency is not considered in the syncytial formation experiments. The vertical axis in Fig 1d uses field as the denominator, which could vary with transfection efficiency. Therefore, I believe that for a more accurate assessment, it is necessary to calibrate with transfection efficiency.
5. In the graph showing the number of viral copies in infected individuals, the vertical axis is linear, so there appears to be a very large difference between GP-C3 and other immuns, but what is actually a fraction of that when compared to vectors?
6. The authors do not state whether the mice showed any weight changes after each vaccination, or whether they developed any other symptoms that might be considered side effects. This point should be included.

Minor comments.

- Line 118. [..., while the monomer Gc level in the GP-2 and GP-C3...] , is this a mistake for GP-C1 and GP-C2? Please verify.
2. Line 179. It would be better for the reader if the main text describes how many days after vaccination the HTNV was challenged.
 3. Line 196. Fig 1f seems to be incorrect, is it correctly Fig 2? Please verify.
 4. Line 202. Fig 2c seems to be incorrect, is it correctly Fig S3g? Please confirm.
 5. Line 205. BUN, Fig S3i? ALB, Fig S3h?
 6. Line 606. mRNA-LNP seems to be incorrect, is it correctly mRNA-LNP?
 7. The GL7 in Fig 6e is too faint and no difference can be seen in print.
 8. Normal values for blood test (cell counts, liver enzymes and renal function) are listed in Fig S3e, but it would be easier to read if each bar graph was shaded or lined to indicate normal values.

RESPONSE TO THE COMMENTS OF REVIEWER

Reviewer #1 (Remarks to the Author):

HFRS caused by Old World hantaviruses leads to severe disease in humans. The inactivated vaccine currently used in endemic areas has limited efficacy, and the development of a new vaccine platform is desired. The authors introduced one to three cysteine mutations in the GP of the Hantaan virus to stabilise it in the pre-fusion state, so that the Gn and Gc of the same molecule, and /or the Gc-Gc of two molecules, are S-S linked. Mice were immunised with the DNA plasmids and/or the mRNA-encapsulated LNPs expressing these GP mutants. Mutant with S-S binding between Gn and Gc of the same molecule and between Gc-Gc of two molecules (called GP-C3) induced the strongest neutralising antibodies in all immunisation experiments. HTNV challenge experiments were also conducted in post-

immunisation mice, showing that the viral load in various organs was reduced and blood markers tested were close to normal. In addition, as there are many inactivated vaccine recipients in the actual endemic area, this vaccine candidates were administered after immunisation with inactivated vaccine, and strong neutralising antibody production was observed. Furthermore, the study also showed that the lymph nodes of the immunised mice had more germinal centre B cells than others, supporting the immunological superiority of this vaccine platform.

This vaccine design appears to be very useful in the prevention of HTNV infection, but there are several points that need to be addressed. It also has a number of typos, which should be checked carefully again.

Response to the comments of Reviewer #1:

We sincerely thank you for your positive assessment of our work and the valuable feedback. We greatly appreciate your constructive suggestions that help us improve our manuscript. We have carefully considered all comments and have revised the manuscript accordingly. Our point-by-point responses to the specific points are detailed below.

Major comments.

Question 1:

1. The histopathology is shown, but it is difficult to determine the degree of inflammation without uninfected individual controls. Uninfected histopathology should also be presented. Were the lungs inflated at necropsy? Also, please provide these pathology images at lowest magnification, as a supporting file. This is to deny that the images presented are the intended extraction of the best parts by the authors.

Response:

Thank you for this important comment regarding the histopathological analysis. We would first like to provide context regarding the animal model. For Old World hantaviruses like HTNV, an ideal animal model that fully recapitulates human disease is lacking. In the standard murine challenge model we employed, HTNV infection is typically subclinical and non-lethal, resulting in inherently modest histopathological changes even in unvaccinated controls. The primary endpoint of this model is the quantification of early viral replication, and observable tissue injury is often subtle. Nevertheless, comparative analysis between groups remains valid and meaningful. As you rightly point out, including uninfected controls is crucial to baseline these subtle changes and to rule out procedure-related artifacts.

In direct response to your suggestions, we have comprehensively revised our

histopathological data presentation.

Noninfected Controls: We have now included histopathological images from uninfected, age-matched control mice in the relevant figures (e.g., revised Fig. 2h,i). These controls confirm the absence of significant background pathology and validate our experimental procedures.

Tissue Processing: Yes, lungs were inflated with formalin at necropsy to preserve alveolar architecture for histological assessment.

Image Presentation: We have reorganized all histopathology figures to address the concern about selective field representation. Each panel now includes a low-magnification panoramic view of the tissue section in the lower-left corner, with a corresponding higher-magnification view of a representative area.

All histopathology figures in the manuscript have been updated accordingly to this new, more transparent format. The representative example from the revised Fig. 2h-i is referenced in our response.

Furthermore, in accordance with the HE staining results, we have also updated the HTNV NP staining images in the manuscript by replacing them with the corresponding lowest-magnification views. An example from the revised **Fig. 2c** is provided below for your reference:

Question 2:

2. More details on the design and structure of the mRNA vaccine are needed, including a clear description of the Cap structure and other RNA modifications.

Response:

Thank you for this suggestion. In response, we have now included a new panel as updated **Fig. 4a** that schematically illustrates the key structural features of our mRNA vaccine construct.

As detailed in the revised manuscript, the mRNA was synthesized with a canonical 5' cap structure (Cap1) to ensure efficient translation initiation. The open reading frame encoding the target antigen is flanked by optimized 5' and 3' untranslated regions (UTRs). Furthermore, to enhance the stability and translational efficiency of the mRNA while reducing its immunogenicity, all nucleosides were modified by the replacement of uridine with N1-methylpseudourine (m1Ψ). We hope these clarifications and the accompanying schematic adequately address the reviewer's inquiry.

While a comprehensive panel summarizing the key RNA modifications and overall mRNA architecture is provided in **Extended Data Fig. 6a**.

Question 3:

3. The WB image in Fig 1C shows the position of (Gn-Gc)x4, but the separation in this region is not good. Doublets are also observed. Please show (Gn-Gc)x4 using a gel with good separation in this region.

Response:

Thank you for raising this valid concern regarding the separation of the (Gn-Gc)x4 complex in the Western blot (Fig. 1c). We acknowledge the suboptimal resolution and presence of doublet bands in that specific gel.

We experienced considerable technical difficulty in achieving clean separation for this high-molecular-weight complex, as the low-percentage PAGE gels required are exceedingly fragile and prone to breakage. Despite extensive optimization attempts, this artifact proved difficult to eliminate completely. The image presented in the revised Fig. 1c represents the best result from these repeated experiments, where the gel remained intact and the multimeric band was clearly discernible.

For full transparency, we provide below several additional independent replicates that consistently show the (Gn-Gc)x4 band (albeit with similar resolution constraints), confirming the reproducibility of multimer formation.

Regarding the additional bands observed, it is important to note that our detection system utilized an antibody against the C-terminal Myc tag engineered on the Gc subunit. Therefore, only protein fragments that retain this C-terminal portion of Gc are visualized. This explains our specific banding pattern, which likely represents common degradation products

or processing intermediates containing the Gc subunit, and differs from patterns obtained with anti-Gn antibodies. This phenomenon of complex banding patterns has been documented in prior studies of hantavirus glycoprotein expression (e.g., *J. Virol.*, 2010, PMID: 19828613).

[Redacted]

We hope the provided explanation and additional data satisfactorily address this point.

Question 4:

4. Transfection efficiency is not considered in the syncytial formation experiments. The vertical axis in Fig 1d uses field as the denominator, which could vary with transfection efficiency. Therefore, I believe that for a more accurate assessment, it is necessary to calibrate with transfection efficiency.

Response:

Thank you for this insightful comment regarding the potential influence of transfection efficiency on our syncytium formation assay.

You rightly point out that transfection efficiency could theoretically influence the count of syncytia per field. In our original experimental setup, we employed a highly standardized and

optimized transfection protocol using a consistent and excess amount of DNA. Under these saturated transfection conditions, the formation of syncytia is primarily governed by the intrinsic fusogenic activity of the expressed glycoproteins, rather than by minor variations in the number of transfected cells. This is why normalizing to the number of visual fields is a well-established practice in such quantitative fusion assays.

To directly address your concern and provide experimental validation, we performed an additional calibration experiment. We transfected 293T cells with a GFP-expressing plasmid under identical conditions and measured transfection efficiency by flow cytometry at multiple time points. The results confirmed that transfection efficiency was consistently high and reached near-maximum levels by 24 hours, with minimal variation. Most importantly, we found no significant correlation between transfection efficiency and syncytium counts across different glycoprotein constructs.

Therefore, we consider the variation in transfection efficiency under our specific conditions to be negligible for the assay readout. These calibration data have been included as **Extended Data Fig. 1d**, and we have updated the manuscript text accordingly. We believe this approach fully addresses your point.

Question 5:

5. In the graph showing the number of viral copies in infected individuals, the vertical axis is linear, so there appears to be a very large difference between GP-C3 and other immuns, but what is actually a fraction of that when compared to vectors?

Response:

Thank you for this insightful observation regarding the interpretation of the viral copy number data. We agree that a linear axis can accentuate visual differences.

To clarify, the statistical comparisons you inquired about were integral to our original analysis of datasets like the one in Fig. 2a. Our methodology (ANOVA with Tukey's test) was applied to assess differences between all relevant groups. The presentation in the figure follows the standard practice of annotating only statistically significant comparisons to maintain clarity. Notably, the lack of significance markers between some prefusion-stabilized groups (e.g., GP-C3 vs. GP-C1/C2) reflects that these specific differences did not reach statistical significance in our analysis, despite numerical trends.

To directly and quantitatively address your question about the "fraction compared to vectors," we provide the exact relative values from our complete dataset below. The vector (for DNA) or LNP (for mRNA) control level is normalized to 1.000 for each organ.

DNA Vaccine Platform:

The table lists all experimental groups' viral load as a fraction of the Vector control in four organs. For example, in the lung (Fig. 2a), GP-C3 is at 0.0924 (9.24%) of the Vector level, while InaVac is at 0.6018 (60.18%). This clearly quantifies the visual difference seen on the linear axis.

Fig. 2a	Ina Vac	Vector	WT	WAASR	GP-C1	GP-C2	GP-C3
As Percentage of Vector	60.18%	100.00%	62.47%	79.15%	15.39%	18.43%	9.24%
Fold Reduction vs. Vector	1.7-fold		1.6-fold	1.3-fold	6.5-fold	5.4-fold	10.8-fold
Fig. 2b	Ina Vac	Vector	WT	WAASR	GP-C1	GP-C2	GP-C3
As Percentage of Vector	61.60%	100.00%	45.45%	41.57%	19.21%	11.56%	13.81%
Fold Reduction vs. Vector	1.6-fold		2.2-fold	2.4-fold	5.2-fold	8.7-fold	7.2-fold
Extended data Fig. 3a	Ina Vac	Vector	WT	WAASR	GP-C1	GP-C2	GP-C3
As Percentage of Vector	50.70%	100.00%	30.51%	13.50%	16.32%	15.88%	10.54%
Fold Reduction vs. Vector	2.0-fold		3.3-fold	7.4-fold	6.1-fold	6.3-fold	9.5-fold
Extended data Fig. 3b	Ina Vac	Vector	WT	WAASR	GP-C1	GP-C2	GP-C3
As Percentage of Vector	16.66%	100.00%	28.54%	23.4%	6.84%	8.48%	6.77%
Fold Reduction vs. Vector	6.0-fold		3.5-fold	4.3-fold	14.6-fold	11.8-fold	14.8-fold

mRNA-LNP Vaccine Platform:

Similarly, the table shows that GP-C3 reduces the viral load to 0.0859 (8.59%) of the

LNP control in the lung (Fig. 5a), demonstrating an even more pronounced effect than its DNA counterpart and significantly surpassing the wild-type GP antigen.

Fig. 5a	Ina Vac	LNP	10 µg GP-WT	10 µg GP-C3
As Percentage of LNP	46.52%	100.00%	23.38%	8.59%
Fold Reduction vs. LNP	2.1-fold		4.3-fold	11.6-fold
Fig. 5b	Ina Vac	LNP	10 µg GP-WT	10 µg GP-C3
As Percentage of LNP	35.31%	100.00%	37.52%	13.98%
Fold Reduction vs. LNP	2.8-fold		2.7-fold	7.2-fold
Extended data Fig. 7a	Ina Vac	LNP	10 µg GP-WT	10 µg GP-C3
As Percentage of LNP	11.34%	100.00%	15.35%	6.79%
Fold Reduction vs. LNP	8.8-fold		6.5-fold	14.7-fold
Extended data Fig. 7b	Ina Vac	LNP	10 µg GP-WT	10 µg GP-C3
As Percentage of LNP	80.57%	100.00%	72.51%	52.22%
Fold Reduction vs. LNP	1.2-fold		1.4-fold	1.9-fold

The full statistical analysis table for Fig. 2a is provided for your reference, which details all pairwise comparisons and P-values.

Statistical Summary for Viral Load in Lung (Fig. 2a)

ANOVA summary	
F	29.50
P value	<0.0001
P value summary	****
Significant diff. among means (P < 0.05)?	Yes
R square	0.8349
Brown-Forsythe test	
F (DFn, DFd)	3.155 (6, 35)
P value	0.0140
P value summary	*
Are SDs significantly different (P < 0.05)?	Yes
Bartlett's test	
Bartlett's statistic (corrected)	51.26
P value	<0.0001
P value summary	****
Are SDs significantly different (P < 0.05)?	Yes

Tukey's multiple comparisons test	Mean Diff.	95.00% CI of diff.	Significant?	Summary	Adjusted P Value
Ina Vac vs. Vector	-2312	-3978 to -645.6	Yes	**	0.0021
Ina Vac vs. WT	-133.2	-1799 to 1533	No	ns	>0.9999
Ina Vac vs. WAASR	-1101	-2768 to 564.9	No	ns	0.3940
Ina Vac vs. GP-C1	2600	933.6 to 4266	Yes	***	0.0004
Ina Vac vs. GP-C2	2423	757.1 to 4090	Yes	**	0.0011
Ina Vac vs. GP-C3	2957	1291 to 4623	Yes	****	<0.0001
Vector vs. WT	2179	512.4 to 3845	Yes	**	0.0041
Vector vs. WAASR	1211	-455.7 to 2877	No	ns	0.2861
Vector vs. GP-C1	4912	3245 to 6578	Yes	****	<0.0001
Vector vs. GP-C2	4735	3069 to 6401	Yes	****	<0.0001
Vector vs. GP-C3	5269	3603 to 6935	Yes	****	<0.0001
WT vs. WAASR	-968.1	-2634 to 698.1	No	ns	0.5466
WT vs. GP-C1	2733	1067 to 4399	Yes	***	0.0002
WT vs. GP-C2	2557	890.3 to 4223	Yes	***	0.0005
WT vs. GP-C3	3090	1424 to 4757	Yes	****	<0.0001
WAASR vs. GP-C1	3701	2035 to 5367	Yes	****	<0.0001
WAASR vs. GP-C2	3525	1858 to 5191	Yes	****	<0.0001
WAASR vs. GP-C3	4058	2392 to 5725	Yes	****	<0.0001
GP-C1 vs. GP-C2	-176.5	-1843 to 1490	No	ns	0.9999
GP-C1 vs. GP-C3	357.3	-1309 to 2023	No	ns	0.9934
GP-C2 vs. GP-C3	533.8	-1132 to 2200	No	ns	0.9503

Question 6:

6. The authors do not state whether the mice showed any weight changes after each vaccination, or whether they developed any other symptoms that might be considered side effects. This point should be included.

Response:

Thank you for raising this important point regarding vaccine safety monitoring. In our study, while daily body weight measurements were not systematically recorded, we performed careful clinical observations of the mice following each vaccination.

Throughout the post-immunization period, all mice were monitored for general health indicators, including activity levels, grooming behavior, posture, and food and water consumption. No overt signs of distress, lethargy, or other adverse effects were observed in any of the immunized groups. The animals remained active and healthy, with no apparent symptoms that would indicate significant vaccine-related side effects.

As you suggested, we have now explicitly included a description of these clinical observations in the revised manuscript in the Results section to provide a more comprehensive safety profile. The specific addition is as follows:

[Throughout the study, all immunized mice remained clinically healthy. Daily observations revealed no signs of distress, lethargy, or changes in grooming, posture, or consumption of food and water following any vaccination.]

We would also like to provide contextual clarification regarding the animal model. The standard murine challenge model used in HTNV vaccine studies involves an acute, non-lethal infection where the virus is typically cleared within days. This model is primarily used to evaluate the vaccine's ability to reduce early viral replication (as measured by tissue viral load), not to model chronic disease or clinical illness. Therefore, monitoring for overt clinical signs or significant weight loss—common endpoints in models of symptomatic viral disease—is not a standard practice or primary focus in this specific field. Our study design, including the clinical observations reported above, aligns with the established endpoints for assessing HTNV vaccine candidates in this model.

Minor comments.

Response to Minor Comments:

We sincerely thank the reviewer for their meticulous reading of our manuscript and for providing these valuable comments to help us improve the clarity and accuracy of our work. We have carefully addressed each point as detailed below. All suggested changes have been incorporated into the revised manuscript.

Question 7:

Line 118. 「..., while the monomer Gc level in the GP-2 and GP-C3...」, is this a mistake for GP-C1 and GP-C2? Please verify.

Response:

Thank you for pointing out this inconsistency. This was a typographical error. The sentence should refer to GP-C1 and GP-C2. We have corrected this in the revised manuscript.

Question 8:

2. Line 179. It would be better for the reader if the main text describes how many days after vaccination the HTNV was challenged.

Response:

Thank you for this suggestion to improve clarity. We have now added the specific information to the main text on original line 179 (line 172). The sentence now reads: ".. HTNV strain 76-118 via intramuscular (i.m.) injection two weeks after the final immunization...." to provide clarity for the reader.

Question 9:

3. Line 196. Fig 1f seems to be incorrect, is it correctly Fig 2? Please verify.

Question 10:

4. Line 202. Fig 2c seems to be incorrect, is it correctly Fig S3g? Please confirm.

Question 11:

5. Line 205. BUN, Fig S3i? ALB, Fig S3h?

Response to Question 9, 10, & 11:

We sincerely apologize for these incorrect figure citations. The reviewer's identifications are accurate.

The citation for "Fig 1f" on line 196 has been corrected to "Fig 2f".

The citation for "Fig 2c" on line 202 has been corrected to "Extended Data Fig. 3g".

On line 205, "BUN" is now correctly cited as "Extended Data Fig. 3i" and "ALB" as "Extended Data Fig. 3h".

We have thoroughly re-checked all figure citations throughout the manuscript to prevent such errors.

Question 12:

6. Line 606. mRNA-LNP seems to be incorrect, is it correctly mRNA-LNP?

Response:

Thank you for noting this typo. This is correct and has been changed to mRNA-LNP throughout the manuscript, including at Line 606.

Question 13:

7. The GL7 in Fig 6e is too faint and no difference can be seen in print.

Response:

We thank you for this important feedback on the figure's clarity. We have reprocessed the original flow cytometry data for Fig. 6e to enhance the contrast and visibility of the GL7 staining, ensuring that the differences between groups are clearly discernible in both digital and print formats. The updated figure has been replaced in the manuscript.

Question 14:

8. Normal values for blood test (cell counts, liver enzymes and renal function) are listed in Fig S3e, but it would be easier to read if each bar graph was shaded or lined to indicate normal values.

Response:

We thank you for this excellent suggestion. In response, we have re-plotted all relevant figures by adding shaded areas or lines to indicate the normal value ranges, which significantly improves readability.

Accordingly, we have revised all relevant bar graphs by adding shaded regions or lines to denote the established normal value ranges for each parameter. This update has been applied to the following figures: Fig. 2d-g, Fig. 5d-h, Extended Data Fig. 3h-j, and Extended Data Fig. 7h-j.

Below is the revised Fig. 2d-g for your reference.

REVIEWER COMMENTS

Reviewer #2 (Remarks to the Author):

The manuscript by Ye and coworkers reports the findings on development of a rationally designed orthohantavirus vaccine prototype. The rationale behind the study is that the vaccine currently used in the HTNV endemic regions, Hantavax, does not necessarily provide a long-lasting strong neutralizing antibody response. Hantavax, to my knowledge, is formalin-inactivated virus (the authors might perhaps include some details of the inactivation method for the vaccine?), which might lead to masking of epitopes essential for neutralization. To tackle the potential instability issues of the spike complex, the authors introduced cysteine residues to selected (based on the 3D structures of the spike complex) regions in the Gn and Gc glycoproteins. The introduction of two disulphide bridges indeed led to stabilization of the heterotetrameric HTNV spike complex based on SDS-PAGE migration, whereas introduction of a single bridge did not act as efficiently.

The authors then went on to test the immunogenicity of the engineered prefusion-stabilized spike in mice using both DNA and mRNA transfection. The extensive studies performed provided evidence supporting the authors' initial hypothesis that prefusion-stabilized spike complex induced more robust neutralizing antibody response than the response resulting from immunization using inactivated virus. In general, the results seem convincing.

Some specific comments/suggestions to improve the manuscript:

-A round of language polishing might be beneficial

-Figure 1a, would it be possible to include similar schematic figures for all constructs tested (at least GP-C1, GP-C2, GP-C3)? This might help the reader to more easily appreciate the differences?

-Have the authors considered of presenting the DNA and mRNA immunization data in the same graphs? This might allow the authors to shorten the manuscript and might allow comparison between the plasmid and mRNA immunizations.

In summary, the authors present interesting data which in my opinion are based on a solid hypothesis (approach utilized in other vaccines, as the authors point out). The conclusions and claims presented appear to be supported by the experimental data. The methods fit well for the work conducted and they have been described in adequate detail.

RESPONSE TO THE COMMENTS OF REVIEWER

Reviewer #2 (Remarks to the Author):

The manuscript by Ye and coworkers reports the findings on development of a rationally

designed orthohantavirus vaccine prototype. The rationale behind the study is that the vaccine currently used in the HTNV endemic regions, Hantavax, does not necessarily provide a long-lasting strong neutralizing antibody response. Hantavax, to my knowledge, is formalin-inactivated virus (the authors might perhaps include some details of the inactivation method for the vaccine?), which might lead to masking of epitopes essential for neutralization. To tackle the potential instability issues of the spike complex, the authors introduced cysteine residues to selected (based on the 3D structures of the spike complex) regions in the Gn and Gc glycoproteins. The introduction of two disulphide bridges indeed led to stabilization of the heterotetrameric HTNV spike complex based on SDS-PAGE migration, whereas introduction of a single bridge did not act as efficiently.

The authors then went on to test the immunogenicity of the engineered prefusion-stabilized spike in mice using both DNA and mRNA transfection. The extensive studies performed provided evidence supporting the authors' initial hypothesis that prefusion-stabilized spike complex induced more robust neutralizing antibody response than the response resulting from immunization using inactivated virus. In general, the results seem convincing.

In summary, the authors present interesting data which in my opinion are based on a solid hypothesis (approach utilized in other vaccines, as the authors point out). The conclusions and claims presented appear to be supported by the experimental data. The methods fit well for the work conducted and they have been described in adequate detail.

Response to the comments of Reviewer #2:

We are deeply grateful for your positive and encouraging assessment of our work. We sincerely appreciate your remarks that our “data are based on a solid hypothesis,” “conclusions are supported by experimental data,” and “methods are well-described.”

Thank you also for raising the point regarding the inactivated vaccine. You are correct in noting that “Hantavax,” the inactivated vaccine currently used in South Korea, is a formalin-inactivated preparation derived from suckling mouse brain, as detailed in the literature (e.g., Vaccine 1999).

[Redacted]

For clarity in our study: the inactivated vaccine (InaVac) used as the control represents the current standard in China. It is a bivalent vaccine (HTNV/SEOV) produced according to the national pharmacopoeia (*Pharmacopoeia of the People's Republic of China, 2020 Edition*), manufactured by culturing the virus in Vero cells and inactivated using **β -Propiolactone**, not formalin. We have clarified this description in the revised manuscript (*Methods section, "Vaccines"*).

[Redacted]

Furthermore, as suggested, we have added a brief background note in the Introduction to provide context: "In East Asia, licensed inactivated vaccines include the mouse brain-derived, formalin-inactivated monovalent HTNV vaccine (Hantavax™) used in South Korea, and bivalent, cell culture-based, beta-propiolactone-inactivated vaccines against both HTNV and SEOV used in China."

We believe these revisions accurately reflect the vaccine landscape and clearly specify the control vaccine used in our study.

Some specific comments/suggestions to improve the manuscript:

Question 1:

-A round of language polishing might be beneficial

Response:

Thank you for this valuable suggestion. We agree that clear and polished language is crucial for scientific communication. In response, the entire manuscript has undergone thorough professional English language editing by a native English-speaking expert in the life sciences. This process has focused on grammar, syntax, word choice, and overall readability to ensure the clarity and precision of our presentation. We are confident the revised manuscript meets the high language standards expected for publication.

Question 2:

-Figure 1a, would it be possible to include similar schematic figures for all constructs tested (at least GP-C1, GP-C2, GP-C3)? This might help the reader to more easily appreciate the differences?

Response:

Thank you for this excellent suggestion. We agree that a clear visual comparison of all constructs will greatly aid the reader. In response, we have now created a new, comprehensive schematic figure (now presented as Extended Data Figure 1a) that includes detailed illustrations of the GP, GP-C1, GP-C2, and GP-C3 constructs, explicitly highlighting the positions of the introduced cysteine pairs for pre-fusion stabilization.

The updated figure is provided below for your reference.

Question 3:

-Have the authors considered of presenting the DNA and mRNA immunization data in the same graphs? This might allow the authors to shorten the manuscript and might allow comparison between the plasmid and mRNA immunizations.

Response:

Thank you for this constructive suggestion. We agree that a direct platform comparison is of great interest. However, the prime-boost schedules and sampling timepoints for the DNA and mRNA regimens were fundamentally different, as each was optimized separately. Plotting these on a shared timeline would be challenging and potentially misleading.

To effectively address your point and facilitate a meaningful comparison, we have added a new Supplementary Table 1. This table consolidates the key immunogenicity and efficacy results (e.g., peak neutralizing antibody titers) for the lead candidates from both platforms, allowing for a clear, side-by-side evaluation of their performance.

To incorporate this direct comparison into the manuscript, we have included the following table in the supplementary information:

Supplementary Table 1. Comparative immunogenicity and efficacy of the prefusion-stabilized GP-C3 vaccine delivered via DNA or mRNA-LNP platforms.

DNA platform	Neutralizing Antibody Titer (GMT) (Day 70)		mRNA-LNP platforms	Neutralizing Antibody Titer (GMT) (Day 28)	
	HTNV	SEOV		HTNV	SEOV
Fig. 1g, h			Fig. 5f, g		
Ina Vac	46.44	40.54	Ina Vac	65.32	44.87
Vector	21.36	25.92	LNP	23.50	25.88
WT	62.64	49.30	2 µg WT	71.84	
GP-C1	54.93	49.54	10 µg WT	84.68	130.39
GP-C2	61.21	62.75	2 µg GP-C3	100.36	
GP-C3	170.12	94.18	10 µg GP-C3	247.16	242.64

	Fold Reduction to Vector			Fold Reduction to LNP	
Fig. 2a, b	Lung	Liver	Fig. 5a, b	Lung	Liver
Ina Vac	1.7-fold	1.6-fold	Ina Vac	2.1-fold	2.8-fold
WT	1.6-fold	2.2-fold			
GP-C1	6.5-fold	5.2-fold	10 µg WT	4.3-fold	2.7-fold
GP-C2	5.4-fold	8.7-fold			
GP-C3	10.8-fold	7.2-fold	10 µg GP-C3	11.6-fold	7.2-fold
	Average Frequency (%)			Average Frequency (%)	
Fig. 4c,e	Tfh	GC B	Fig. 5a, b	Tfh	GC B
Ina Vac	7.61	3.8	Ina Vac	7.80	6.22
Vector	1.48	1.7	LNP	3.19	3.85
WT	4.08	4.8	10 µg WT	10.02	6.32
GP-C3	14.61	9.52	10 µg GP-C3	26.79	14.74

We believe this approach provides an accurate and useful comparison while preserving the clarity of the individual kinetic data in the main figures.

REVIEWER COMMENTS

Reviewer #3 (Remarks to the Author):

Ye et al. describe the development and testing of prefusion-stabilized Hantaan virus (HTNV) vaccines. The study involved the rational design of DNA and mRNA-LNP vaccines, as well as the evaluation of their immunogenicity and protective efficacy in mice. The vaccine formulations were analyzed for their biochemical and histopathological effects, as well as their ability to induce humoral and cellular immune responses. The study also included the measurement of viral burden in challenged mouse tissues and the assessment of serological parameters. The results demonstrated that the prefusion-stabilized HTNV GP-based nucleic acid vaccines (especially the GP-C3 one) elicited robust humoral and cellular immunity, providing potential protection against HTNV-induced histopathological injury and other pathological effects. The authors did an extensive experimental work on selecting the best vaccine regarding the one they tested, however important majors and minors points need to be addressed before considering publishing this study.

Major points

The authors have already contributed significantly to the field with their previous publications on the development of an HTNV VSV-based vaccine (npj vaccines PMID: 38341504) and Hantaan virus pathogenicity (Nature Comm PMID: 38200007). Additionally, another preprint article coming from the same University and investigating glycoprotein-based nucleic acid vaccination for the effective control of HTNV epidemics is under review at npj vaccines (<https://doi.org/10.21203/rs.3.rs-3933421/v1>). However, it's worth noting that the present study shares considerable similarity with these 2024 HTNV vaccine studies, potentially impacting the novelty of the current work.

One important limitation of the study, as acknowledged by the authors themselves, is the significant lack of evaluation of GP-specific B cells. The stability of GP-C3 and GP-C2 proteins allows for their characterization by SDS-PAGE +/- β -mercaptoethanol. However, employing affinity chromatography coupled with fPLC purification could potentially enabling the recovery of GP-C3 and GP-C2 proteins, for further linkage to fluorophores, enabling specific B-cell sorting. Another possibility would be using Gn/Gc construct to make flow cytometry probes to sort GP-specific B cells. Evaluating affinity maturation of sorted B-cells could offer valuable information regarding the immunological response, including VH antibody classes, sequences, and mutations needed for effective neutralization of the virus and protection against HTNV infection. Regarding this study, while the authors have evaluated GC B cells numbers in LNs, information about the presence of GCs at d70 or d180 post-immunization (instead of only after 7 days) and the role of GP-C3 DNA immunogen in inducing a significant humoral memory response for long-term protection against HTNV is lacking.

Furthermore, the paper evaluate both DNA vaccine and RNA LNP vaccine studies, which are presented in a manner that appears as two separate studies within a single article. Restructuring the presentation to focus on comparing GP-C3 as DNA or RNA vaccines would enhance the coherence of the narrative and aid in identifying the most effective HTNV GP vaccine candidate.

Minor points:

- Legends for Fig 1c (electrophoresis) are different from the text (line 112,113);
- Why using two different plasmids for HTNV transfection?
- Line 118, GP-C2 and GP-C1 (instead of GP-C3);
- Line 121, "Fig S1b" is maybe Fig S1c?
- Line 124, "Fig S1c" is the electrophoresis figure, not the syncytia. Maybe Fig S1b?

- For the neutralization assay of SEOV virus, why did you choose to only take d14 sera and not d70? How the SEOV Nab levels induced by stabilized GP-C1 and GP-C2 are lesser than those induced by the attenuated Ina Vac (Fig S1d)?

- Why scarifying mice after just 3 days of challenge? We generally wait for 5 days to do a post-challenge assess virus dissemination (PMID: 32667279). It would also be better to follow the wight loss for 14 days to see any sign of infection and/or protection.

- In Figure 2 and figure S3, a statistical fluorecence evaluation is need as the GP-C3 seems to be significantly lowering the viral load only in the kidney, compared to attenuated Ina Vac and WT.
- Line 196, why "Fig 1f"?
- Statistical analysis in all graphs are confusing. To make it more powerful, it would be better to have results interpretation comparing the DNA immunogens to the attenuated Ina Vac or WT, and not to the vector.

- A healthy cotrol (non-infected) for all tissue histology is missing in the comparison.

- Line 234, figure "1h" is showing IFN γ SFC, you mean "Fig 1g"
- Line 292, is "4d"
- Line 295, this sentence is not true as it is now well known that one of the weakest points of mRNA vaccines, especially SARS-CoV2 mRNA vaccines is that they lack on effective long term T-cell activation.
- One of the interesting parts of this study is the mixed vaccine protocol by priming with the inactivate Ina Vac and boosting with different GP construct. This part, in my opinion, is the

most innovative and should've received more focus for the immune response elicited and to suggest a better way to target the immune system to a more specific and protecting HTNV immune response.

Overall, while the study presents intriguing findings, addressing the aforementioned concerns and enhancing experimental clarity and coherence is significantly needed and would substantially improve the manuscript to be considered for a publication in Nature Communications.

RESPONSE TO THE COMMENTS OF REVIEWER

Reviewer #3 (Remarks to the Author):

Ye et al. describe the development and testing of prefusion-stabilized Hantaan virus (HTNV) vaccines. The study involved the rational design of DNA and mRNA-LNP vaccines, as well as the evaluation of their immunogenicity and protective efficacy in mice. The vaccine formulations were analyzed for their biochemical and histopathological effects, as well as their ability to induce humoral and cellular immune responses. The study also included the measurement of viral burden in challenged mouse tissues and the assessment of serological parameters. The results demonstrated that the prefusion-stabilized HTNV GP-based nucleic acid vaccines (especially the GP-C3 one) elicited robust humoral and cellular immunity, providing potential protection against HTNV-induced histopathological injury and other pathological effects. The authors did an extensive experimental work on selecting the best vaccine regarding the one they tested, however important majors and minors points need to be addressed before considering publishing this study.

Overall, while the study presents intriguing findings, addressing the aforementioned concerns and enhancing experimental clarity and coherence is significantly needed and would substantially improve the manuscript to be considered for a publication in Nature Communications.

Response to the comments of Reviewer #3:

We would like to express our sincere gratitude for your thorough evaluation of our manuscript and for providing these insightful and constructive comments. We have carefully considered each point and, where applicable, performed additional experiments and analyses to address them. We believe these revisions have substantially strengthened our study, and we are grateful for the time and expertise you have dedicated to improving our work. Our point-by-point responses are detailed below.

Major points

Question 1:

The authors have already contributed significantly to the field with their previous publications on the development of an HTNV VSV-based vaccine (npj vaccines PMID: 38341504) and Hantaan virus pathogenicity (Nature Comm PMID: 38200007). Additionally, another preprint article coming from the same University and investigating glycoprotein-based nucleic acid vaccination for the effective control of HTNV epidemics is under review at npj vaccines (<https://doi.org/10.21203/rs.3.rs-3933421/v1>). However, it's worth noting that the present study shares considerable similarity with these 2024 HTNV vaccine studies, potentially impacting the novelty of the current work.

Response:

We sincerely thank you for your thorough evaluation and for raising this important point regarding the novelty of our work in the context of related studies. We appreciate the opportunity to clarify the distinct focus and innovation of present study.

The cited works are complementary but address fundamentally different scientific questions and employ distinct technological approaches:

(1) VSV-vectored vaccine (npj Vaccines, 2024, PMID: 38341504): Our previous study on the VSV-vectored vaccine (npj Vaccines, 2024), is focus on developing a replicating viral vector vaccine using the unmodified, wild-type HTNV glycoprotein as the antigen. Due to challenges in expressing soluble GP at that time, the detection of HTNV-specific antibodies in that work relied on immunofluorescence assay (IFA). Notably, that study led to the development of a serological method, using purified VSV particles displaying the HTNV GP as the coating antigen based ELISA to detect the HTNV GP antibody level. The present study has adopted this method for the quantitative analysis of antibodies. Thus, the prior work is fundamentally to our present work, but not an overlap in novelty.

(2) Pathogenicity study (Nature Communications, 2024, PMID: 38200007): This work was a mechanistic investigation into virus-host interactions, specifically identifying host factors that influence HTNV pathogenesis. Its primary goal was understanding disease mechanisms, not vaccine development. Therefore, it does not overlap with the preventive vaccine aims of the current manuscript.

(3) Related nucleic acid vaccine study (now published in npj Vaccines 2024, PMID: 39443512): We have carefully examined this study. Our work maintains clear novelty for two key reasons:

a) **Antigen Design:** The core innovation of our manuscript is the structure-guided

rational design of prefusion-stabilized HTNV glycoprotein antigens (GP-C1, GP-C2, GP-C3). To our knowledge, this approach has not been reported elsewhere for hantavirus vaccines, including in the cited npj Vaccines study, which used the unmodified, wild-type GP gene.

b) **mRNA Platform:** Our study presents, to our knowledge, the first report of an LNP-formulated mRNA vaccine encoding the HTNV glycoprotein. The cited study used an LNP for DNA delivery and a polymeric transfection reagent (not LNP) for mRNA delivery, which are formulationally and functionally distinct from our optimized mRNA-LNP platform.

To further support our clarification regarding the distinct mRNA delivery platforms, we provide below two relevant excerpts for your reference:

b-1) Excerpt from the preprint (now published in npj Vaccines): The figure legend clearly states that the mRNA vaccine was delivered using "*in vivo* mRNA transfection reagent (Polyplus)," confirming it was not an LNP-based formulation.

DOI: <https://doi.org/10.21203/rs.3.rs-3933421/v1>

598 Both DNA and DNA-LNP candidate vaccines contained 30 µg of DNA with an
599 injection volume of 50 µL. The mRNA candidate vaccine comprised 10 µg mRNA,
600 which is delivered using *in vivo* mRNA transfection reagent (Polyplus). The injection

b-2) Excerpt from the manufacturer's (Polyplus) product documentation: The description explicitly differentiates their ready-to-use liposomal reagent (in vivo-jetRNA®+) from lipid nanoparticles (LNPs), noting that it "does not require any formulation equipment," which is a key technological distinction from the LNP platform used in our study.

[Redacted]

These materials corroborate our point that the cited study utilized a delivery system fundamentally different from the mRNA-LNP platform central to our work.

We are confident that the rational antigen design coupled with the novel mRNA-LNP delivery presented in our manuscript represents a significant and distinct advance in the field of hantavirus vaccinology. We thank the reviewer again for prompting this important clarification.

The specific addition is as follows:

[This outcome stands in contrast to a recent study in which delivery of the wild-type HTNV glycoprotein via alternative nucleic acid platforms elicited only modest NAb titers, underscoring the critical contribution of our prefusion-stabilization strategy to achieving potent humoral immunity⁶⁶.]

Question 2:

One important limitation of the study, as acknowledged by the authors themselves, is the significant lack of evaluation of GP-specific B cells. The stability of GP-C3 and GP-C2 proteins allows for their characterization by SDS-PAGE +/- β -mercaptoethanol. However, employing affinity chromatography coupled with fPLC purification could potentially enabling the recovery of GP-C3 and GP-C2 proteins, for further linkage to fluorophores, enabling specific B-cell sorting. Another possibility would be using Gn/Gc construct to make flow cytometry probes to sort GP-specific B cells. Evaluating affinity maturation of sorted B-cells could offer valuable information regarding the immunological response, including VH antibody classes, sequences, and mutations needed for effective neutralization of the virus and protection against HTNV infection. Regarding this study, while the authors have evaluated GC B cells numbers in LNs, information about the presence of GCs at d70 or d180 post-immunization (instead of only after 7 days) and the role of GP-C3 DNA immunogen in inducing a significant humoral memory response for long-term protection against HTNV is lacking.

Response:

We sincerely thank you for these insightful suggestions regarding the in-depth analysis of GP-specific B cells and the kinetics of the germinal center (GC) response.

(1) Regarding the evaluation of GP-specific B cells:

We fully appreciate the reviewer's suggestion to use purified pre-fusion stabilized proteins as probes for B cell sorting. Indeed, we have actively pursued this direction. However, our extensive efforts to express and purify the full-length, pre-fusion stabilized HTNV GP have not yet yielded protein of sufficient quality for probe generation, which reflects a recognized challenge in the field (detailed in response to Q3).

Furthermore, while expressing individual subunits (Gn or Gc) is more feasible, evidence suggests they are suboptimal for unbiased B cell sorting, as they likely miss B cell clones targeting complex epitopes on the intact spike. Using such probes might therefore yield incomplete data that does not fully represent the broad neutralizing response induced by our vaccines.

Given these technical considerations, we quantitatively measured GC B cell and T follicular helper (Tfh) cell frequencies as well-established correlates of ongoing B cell activation. Our data on high-titer, protective neutralizing antibodies serve as strong functional readouts of a high-quality B cell response. We have included this suggestion as a key direction for future research in the revised Discussion.

(2) Regarding the timing of GC B cell analysis:

We appreciate your point on the kinetics of the GC responses at later time points (e.g., d70 or d180 post a single immunization) to understand long-lived humoral memory development.

Our decision to analyze lymph nodes at day 7 post-primary immunization was based on

established immunological kinetics and standard practice in the field for evaluating the peak induction of GCs, which is the optimal window to capture and compare the magnitude of the initial B cell response elicited by different vaccine constructs. As supported by numerous studies, GC activity in draining LNs typically peaks within the first 1-2 weeks and subsequently contracts, as consistently demonstrated in the literature (e.g., *J Exp Med*, 2018; *Immunity*, 2020; *Nat Immunol*, 2022). Therefore, later timepoints would not better assess the vaccine's capacity to initiate a robust GC reaction - our primary objective.

The specific addition is as follows:

[To test this, BALB/c mice received a single i.m. dose of the different vaccines. Draining lymph nodes (LNs) were analyzed 7 days post-immunization—a well-established peak for germinal center formation—by flow cytometry seven days post-immunization.]

To specifically address the role of the GP-C3 DNA immunogen in inducing long-term protective memory, our study provides two key lines of functional evidence:

a) **Durability of the Primary Response:** The neutralizing antibody titers elicited by the GP-C3 DNA vaccine itself were maintained at high levels through 180 days post-immunization. This directly demonstrates its capacity to induce a durable humoral response.

b) **Efficacy in Recalling Memory:** In a heterologous prime-boost model, a GP-C3 boost given 180 days after a minimal priming dose elicited a robust anamnestic response, proving its superior ability to recruit and expand pre-existing memory B cells.

While directly tracking GCs at ultra-late time points after a single immunization could add detail, our combination of peak GC/Tfh analysis, sustained antibody titers, and functional memory recall assays provides a comprehensive and compelling assessment of the quality and durability of the immune response elicited by GP-C3.

Below are supporting examples from related studies demonstrating the standard practice of analyzing LN responses at peak time points.

[Redacted]

Question 3:

One important limitation of the study, as acknowledged by the authors themselves, is the significant lack of evaluation of GP-specific B cells. The stability of GP-C3 and GP-C2 proteins allows for their characterization by SDS-PAGE +/- β -mercaptoethanol. However, employing affinity chromatography coupled with fPLC purification could potentially enabling the recovery of GP-C3 and GP-C2 proteins, for further linkage to fluorophores, enabling specific B-cell sorting. Another possibility would be using Gn/Gc construct to make flow cytometry probes to sort GP-specific B cells. Evaluating affinity maturation of sorted B-cells could offer valuable information regarding the immunological response, including VH antibody classes, sequences, and mutations needed for effective neutralization of the virus and protection against HTNV infection.

Response:

We sincerely appreciate this valuable suggestion regarding the evaluation of GP-specific B cells. We agree that analyzing the B cell receptor repertoire at a high resolution would provide a deeper understanding of the immune response elicited by our pre-fusion stabilized antigens. The reviewer's proposed strategies—using purified, stabilized GP proteins (GP-C2/C3) or Gn/Gc constructs as probes for flow cytometry-based B cell sorting—represent the gold-standard approach for this type of investigation. We fully acknowledge the significant value this would add, and we would like to clarify the significant technical challenges associated with generating the necessary probes for such analysis in the context of HTNV.

Producing the requisite high-quality antigens for B cell probing, which currently presents a major bottleneck in the field for HTNV and related viruses. As you may be aware, the expression and purification of the full, pre-fusion HTNV glycoprotein spike with correct conformational integrity is exceptionally difficult. Prior to the seminal 2020 Cell paper (PMID: 32937107), there were no successful reports of producing the near-full-length HTNV GP in a recombinant form suitable for structural or deep immunological studies. A recent protocol from the Pasteur Institute in Methods in Molecular Biology (PMID: 38315356) has detailed their method, underscoring the complexity and specialized expertise required. We have been actively working to implement this exact strategy in our laboratory but have not yet successfully reproduced it to obtain protein of sufficient quality and quantity for probe generation.

While expressing individual subunits (Gn or Gc) is somewhat more feasible, published evidence suggests they are suboptimal for unbiased B cell sorting. For instance, a 2020 mBio study (PMID: 34225492) that used recombinant HTNV Gn to sort B cells from immunized rabbits recovered a relatively low number of neutralizing antibodies. This indicates that probes based on single subunits likely miss a significant fraction of the B cell repertoire, particularly those clones specific for complex, quaternary epitopes present only on the intact

spike—which are often the most potent neutralizers. Like evidence from Sci Transl Med study (PMID: 37315110) which the most potent NAb ADI-42898 is target such epitope.

[Redacted]

Therefore, using a non-optimal probe (like a single subunit) risks generating a skewed and incomplete dataset that may not accurately reflect the broad and potent neutralizing response our GP-C3 vaccine induces, as evidenced by the high serum neutralizing antibody titers.

Given these formidable technical hurdles, we relied on robust surrogate measures of the GC reaction and B cell help—namely, the direct quantification of GC B cells and Tfh cells in the draining LNs at the peak of the response, coupled with the functional endpoint of high-titer, protective neutralizing antibodies in serum. We are confident these data strongly indicate the successful initiation of a potent T cell-dependent B cell response.

We have added a paragraph to the Discussion to explicitly acknowledge this limitation, to thank the reviewer for the suggestion, and to state that overcoming this protein production challenge to enable such high-resolution B cell analysis is a critical and active goal of our ongoing research. We are currently optimizing the expression and purification protocol based on the published methods (PMID: 38315356) to generate the necessary conformational probes for future B cell repertoire studies.

[Finally, technical hurdles in producing full-length, prefusion-stabilized HTNV GP of sufficient quality have so far precluded the generation of probes for detailed GP-specific B cell receptor analysis—a recognized challenge in the field. Nonetheless, our comprehensive functional readouts—high-titer and durable NAbs, robust GC responses, and solid protection against challenge—collectively demonstrate the success of our rational vaccine design strategy.]

Question 4:

Furthermore, the paper evaluate both DNA vaccine and RNA LNP vaccine studies, which are presented in a manner that appears as two separate studies within a single article. Restructuring the presentation to focus on comparing GP-C3 as DNA or RNA vaccines would enhance the coherence of the narrative and aid in identifying the most effective HTNV GP vaccine candidate.

Response:

We thank you for this insightful suggestion regarding the manuscript's structure. We agree that a clearer comparison between the DNA and mRNA-LNP platforms would be valuable.

Upon careful consideration, we determined that presenting the DNA and mRNA data in the same graphs or a completely unified narrative is not technically feasible. This is because the immunization protocols (e.g., prime-boost schedules, dosing, bleeding timepoints) were optimized independently for each platform to reflect their distinct pharmacological properties. Aligning these disparate timelines on a single graph would be misleading and could obscure the platform-specific immune kinetics we aimed to capture.

However, we have fully embraced your core objective—to enable the reader to easily compare the outcomes across platforms. To achieve this, we have implemented the following key changes in the revised manuscript:

1. We have reorganized the Results section to first present the rational design and the initial *in vivo* screening using the DNA platform, which identified GP-C3 as our lead candidate. We then introduce a dedicated subsection that explicitly focuses on evaluating this same lead candidate (GP-C3) in the mRNA-LNP platform.

2. Most importantly, we have created a new Supplementary Table (See Supplementary Table 1) that directly juxtaposes the peak immunogenicity data (e.g., neutralizing antibody titers, GC B cell responses) and protective efficacy for the GP-C3 construct across both the DNA and mRNA-LNP platforms. This allows for a direct, side-by-side, and scientifically accurate comparison of the final performance metrics, which was the ultimate goal of the reviewer's suggestion.

This revised structure starts from rational design yielded an optimal antigen (GP-C3), which proves to be highly immunogenic and protective when delivered by two distinct, clinically relevant nucleic acid platforms. We believe this approach now provides the clarity and comparative insight the reviewer requested, while maintaining the integrity of our individual experimental designs.

Minor points:**Response to Minor Comments:**

We thank you for your meticulous reading and these valuable technical and editorial comments. We have carefully addressed each point below, and all corresponding corrections have been incorporated into the revised manuscript and figures.

Question 5:

- Legends for Fig 1c (electrophoresis) are different from the text (line 112,113);

Response:

We apologize for this inconsistency. The figure legend and the main text have been carefully cross-checked and unified in the revised manuscript.

Question 6:

- Why using two different plasmids for HTNV transfection?

Response:

We thank you for this question, which allows us to clarify a potentially confusing point in our methodology description.

In our study, we utilized two different plasmid backbones for distinct experimental purposes:

1. The pCAGGS plasmid was used for the *in vitro* syncytium formation assay and other initial protein expression validations. This vector possesses a strong promoter that drives high-level transient protein expression, making it ideal for such functional cell-based assays.

2. The pVAX1 plasmid was used for all *in vivo* DNA immunization studies. This vector is a well-established and widely used backbone specifically designed and optimized for DNA vaccine applications, with a strong safety profile.

Therefore, the use of these two different plasmids was a deliberate choice to leverage the specific advantages of each system, pCAGGS for robust *in vitro* analysis and pVAX1 for standardized *in vivo* vaccination. To ensure clarity for all readers, we have incorporated this rationale into the revised manuscript. Specifically, we have added the following explanation in the “Cells, plasmids, viruses” section of the Methods:

[Thus, the pCAGGS vector was utilized for *in vitro* assays requiring high transient expression, while the pVAX1 vector, a backbone optimized for DNA vaccines, was employed for all animal immunization studies.]

Question 7:

- Line 118, GP-C2 and GP-C1 (instead of GP-C3);

Response:

We sincerely apologize for this error. The text on line 118 has been corrected to "GP-C1 and GP-C2".

Question 8:

- Line 121, "Fig S1b" is maybe Fig S1c?

Question 9:

- Line 124, "Fig S1c" is the electrophoresis figure, not the syncytia. Maybe Fig S1b?

Response to Q8 & Q9:

We thank you for identifying these incorrect citations. We have thoroughly re-checked all **Extended Data Fig** citations throughout the manuscript. The citations on lines 121 and 124 (and all others) have been corrected to accurately reference the intended data panels (e.g., "Fig S1b" and "Fig S1c" have been swapped as per the reviewer's correct identification). We have updated citation with **Extended Data Fig. 1c (the electrophoresis figure)** and **Extended Data Fig. 1e (the syncytia formation panel)**.

Question 10:

- For the neutralization assay of SEOV virus, why did you choose to only take d14 sera and not d70? How the SEOV Nab levels induced by stabilized GP-C1 and GP-C2 are lesser than those induced by the attenuated Ina Vac (Fig S1d)?

Response:

Regarding (Part 1): For the neutralization assay of SEOV virus, why did you choose to only take d14 sera and not d70?

Your point is well taken. In the initial submission, we presented the d14 data to provide an early snapshot of cross-neutralizing potential. In response to this comment, we have now performed the neutralization assay using d70 sera from the DNA vaccine groups and included these results as a new panel (**Fig. 1h**; provided below for reference). The results are consistent with the d14 data, showing GP-C3 exhibit an ideal cross-neutralization effect. This provides a more comprehensive view of the durability and evolution of the cross-neutralizing response.

In addition, we have also included the cross-neutralization data against SEOV at 4 weeks post-immunization for the mRNA-LNP vaccine platform as **Fig.4g** (provided below for reference), offering a parallel view of cross-protective potential across both delivery platforms.

Regarding (Part 2): How the SEOV Nab levels induced by stabilized GP-C1 and GP-C2 are lesser than those induced by the attenuated Ina Vac (Fig S1d)?

We appreciate this insightful observation. First, we wish to clarify a key point for accurate interpretation: the inactivated vaccine (Ina-Vac) used as a control in our study is a **bivalent vaccine containing both HTNV and SEOV antigens**, as licensed for use in endemic regions (different from South Korea approved **monovalent** “Hantavax”, detailed in our Response to the comments of Reviewer #2). Therefore, the early (d14) SEOV-neutralizing antibody (NAb) titers it induces are primarily a direct response to its SEOV component.

In this context, the initially lower SEOV NAb levels induced by our monovalent HTNV-derived GP-C1 and GP-C2 constructs are expected, as their early activity relies entirely on cross-reactive B cell clones. The significant increase in their SEOV cross-neutralizing titers from d14 to d70 suggests an ongoing process of affinity maturation, whereby the immune response refines towards conserved, cross-reactive epitopes.

The superior and consistent cross-neutralization induced by GP-C3 across all time points and platforms is the key finding. This can be attributed to the rational design of GP-C3. We hypothesize that its specific disulfide bond combination optimally stabilizes the prefusion conformation while best preserving critical cross-reactive epitopes shared between HTNV and SEOV.

This concept is powerfully supported by the recent discovery of a broad-scale neutralizing human antibody, ADI-42898, which targets a complex, quaternary epitope spanning both Gn and Gc subunits and neutralizes both HTNV and SEOV (Sci Transl Med. PMID: 37315110; the relevant figure panels from this study is indicated in our response to de Q3). The efficacy of ADI-42898 confirms the existence and importance of such conserved, conformation-dependent epitopes—precisely the type of epitopes our GP-C3 design appears to effectively present.

In contrast, the suboptimal cross-neutralization by GP-C1 and GP-C2 suggests their particular stabilization strategies might inadvertently alter some of these key cross-reactive epitopes, despite benefiting homologous HTNV neutralization. Therefore, this comparison underscores that subtle differences in antigen engineering can significantly impact the breadth of the immune response, and GP-C3 represents an optimally balanced design.

We have incorporated discussion of these points, including the reference to ADI-42898, in the revised manuscript.

Question 11:

- Why scarifying mice after just 3 days of challenge? We generally wait for 5 days to do a post-challenge assess virus dissemination (PMID: 32667279). It would also be better to follow the wight loss for 14 days to see any sign of infection and/or protection.

Response:

We thank you for this methodological insight regarding the challenge timepoint. The choice to assess viral load at 3 days post-challenge in our BALB/c mouse model was based directly on the foundational kinetic profile established by our own research group.

Specifically, in our previous study using the C57BL/6 model (Cheng LF, et al. Detection of specific antigens in tissues of C57BL/6 mice infected with Hantaan virus. Journal of Tropical Medicine (in Chinese). 2011; 11(05): 481-482+499), we systematically demonstrated that the viral load in key target organs peaks around day 3 post-infection. While the current study utilizes the BALB/c strain, the well-documented similarities in the general pathogenesis and immune response kinetics between these two common mouse strains provide strong rationale for applying this established, sensitive time point to our present vaccine efficacy study.

Building on this foundational work from our lab, we determined that day 3 post-challenge remains the most suitable window to rigorously assess the ability of our vaccines to reduce peak viral replication.

The key findings from that study are summarized in the table below, reproduced for your reference. It shows the detection of HTNV antigen across various tissues at different days post-infection (d.p.i.):

[Redacted]

To incorporate this important methodological clarification into the manuscript, we have added the following sentence to the revised “DNA vaccine immunization regime” section of the Methods:

[Three days post-infection—corresponding to the peak of viral replication in murine model—tissues were collected for analysis.]

We acknowledge that incorporating clinical monitoring (e.g., weight loss) over an extended period could offer complementary readouts. In our study, while daily body weight measurements were not systematically recorded, we performed careful clinical observations of the mice following each vaccination. Throughout the post-immunization period, all mice were monitored for general health indicators, including activity levels, grooming behavior, posture, and food and water consumption. No overt signs of distress, lethargy, or other adverse effects were observed in any of the immunized groups. The animals remained active and healthy, with no apparent symptoms that would indicate significant vaccine-related side effects.

We would also like to provide contextual clarification regarding the animal model. The standard murine challenge model used in HTNV vaccine studies involves an acute, non-lethal infection where the virus is typically cleared within days. This model is primarily used to evaluate the vaccine’s ability to reduce early viral replication (as measured by tissue viral load), not to model chronic disease or clinical illness. Therefore, monitoring for overt clinical signs or significant weight loss—common endpoints in models of symptomatic viral disease—is not a standard practice or primary focus in this specific field. Our study design, including the clinical observations reported above, aligns with the established endpoints for assessing HTNV vaccine candidates in this model.

Question 12:

- In Figure 2 and figure S3, a statistical fluorescence evaluation is need as the GP-C3 seems to be significantly lowering the viral load only in the kidney, compared to attenuated Ina Vac and WT.

Response:

We thank you for raising this important point regarding the quantitative assessment of viral load in tissues. We agree that a statistical evaluation provides a more objective and robust measure than qualitative observation alone.

In direct response to this comment, we have re-analyzed all *in vivo* immunofluorescence images for viral antigen detection. For each image set, we have now performed a quantitative analysis of the relative fluorescence intensity across multiple fields of view for each group and tissue.

Specifically, we performed a quantitative statistical analysis of the relative fluorescent intensity for all groups across all organs. This was done by measuring the fluorescence intensity in the regions of interest using ImageJ software, normalizing the data to the mean value of the control group (set as 1), and then conducting direct statistical comparisons between the key groups, including GP-C3, InaVac, and WT.

These new quantitative data, accompanied by the corresponding statistical analysis, have been added to the respective figures as new panels (**please refer to the revised Extended data Fig 3c-e and Extended data Fig 7c-e provided below**). The results of this analysis confirm and extend our initial observations, providing a clear statistical basis for the reduction in viral antigen load mediated by the GP-C3 vaccine candidate.

We believe this addition significantly strengthens the evidence presented in the manuscript.

Question 13:

- Line 196, why “Fig 1f”?

Response:

This was an incorrect citation. It has been corrected to the appropriate figure reference in the revised manuscript.

Question 14:

- Statistical analysis in all graphs are confusing. To make it more powerful, it would be better to have results interpretation comparing the DNA immunogens to the attenuated Ina Vac or WT, and not to the vector.

Response:

We thank you for this critical comment regarding the clarity of statistical comparisons.

We would like to clarify that our original statistical analysis already included the comparisons between key experimental groups (e.g., vaccine candidates vs. InaVac, vs. WT) using one-way ANOVA followed by Tukey's multiple comparisons test, as described in our Methods. However, we acknowledge that the presentation of these specific comparisons in the figures could have been clearer.

In response to the reviewer's suggestion, we have revised all relevant figures to make the direct statistical comparisons between key groups (e.g., GP-C3 DNA/mRNA vs. InaVac and vs. WT) more prominent, alongside comparisons against the vector control.

The results text has been refined to focus on these biologically relevant comparisons. We explicitly note that while GP-C3 consistently shows strong immunogenicity, its difference from the inactivated vaccine (InaVac) does not always reach statistical significance for certain endpoints. This nuanced interpretation underscores that our rationally designed antigen can induce protection comparable to the traditional vaccine.

To ensure full transparency, we confirm that the suggested comparisons were integral to our original analysis (One-way/Two-way ANOVA with Tukey's test). An excerpt of this analysis for a representative dataset is provided below for the reviewer's reference.

Statistical Analysis Example (Two-way ANOVA with Tukey's multiple comparisons test):

Fig 1g

Two-way ANOVA	Ordinary				
Alpha	0.05				
Source of Variation	% of total variation	P value	P value summary	Significant?	
Interaction	9.149	0.0015	**	Yes	
Row Factor	5.019	0.0001	***	Yes	
Column Factor	64.52	<0.0001	****	Yes	
ANOVA table	SS	DF	MS	F (DFn, DFd)	P value
Interaction	22732	12	1894	F (12, 84) = 3.005	P=0.0015
Row Factor	12469	2	6235	F (2, 84) = 9.889	P=0.0001
Column Factor	160301	6	26717	F (6, 84) = 42.38	P<0.0001

Residual	52960	84	630.5		
Tukey's multiple comparisons test	Mean Diff.	95.00% CI of diff.	Significant ?	Summary	Adjusted P Value
Day 14					
Ina Vac vs. Vector	35.27	-12.70 to 83.25	No	ns	0.2955
Ina Vac vs. WT	-18.83	-66.80 to 29.15	No	ns	0.8977
Ina Vac vs. WAASR	-8.704	-56.68 to 39.27	No	ns	0.9980
Ina Vac vs. GP-C1	-49.08	-97.06 to -1.106	Yes	*	0.0415
Ina Vac vs. GP-C2	-38.69	-86.66 to 9.284	No	ns	0.1968
Ina Vac vs. GP-C3	-85.66	-133.6 to -37.68	Yes	****	<0.0001
Vector vs. WT	-54.10	-102.1 to -6.129	Yes	*	0.0168
Vector vs. WAASR	-43.98	-91.95 to 3.996	No	ns	0.0943
Vector vs. GP-C1	-84.36	-132.3 to -36.38	Yes	****	<0.0001
Vector vs. GP-C2	-73.96	-121.9 to -25.99	Yes	***	0.0002
Vector vs. GP-C3	-120.9	-168.9 to -72.96	Yes	****	<0.0001
WT vs. WAASR	10.13	-37.85 to 58.10	No	ns	0.9953
WT vs. GP-C1	-30.25	-78.23 to 17.72	No	ns	0.4829
WT vs. GP-C2	-19.86	-67.84 to 28.11	No	ns	0.8719
WT vs. GP-C3	-66.83	-114.8 to -18.85	Yes	**	0.0012
WAASR vs. GP-C1	-40.38	-88.35 to 7.597	No	ns	0.1577
WAASR vs. GP-C2	-29.99	-77.96 to 17.99	No	ns	0.4937
WAASR vs. GP-C3	-76.95	-124.9 to -28.98	Yes	***	0.0001
GP-C1 vs. GP-C2	10.39	-37.58 to 58.36	No	ns	0.9946
GP-C1 vs. GP-C3	-36.58	-84.55 to 11.40	No	ns	0.2548
GP-C2 vs. GP-C3	-46.97	-94.94 to 1.008	No	ns	0.0590
Day 42					
Ina Vac vs. Vector	100.5	52.53 to 148.5	Yes	****	<0.0001
Ina Vac vs. WT	40.23	-7.741 to 88.21	No	ns	0.1608
Ina Vac vs. WAASR	46.48	-1.496 to 94.45	No	ns	0.0639
Ina Vac vs. GP-C1	53.62	5.647 to 101.6	Yes	*	0.0184
Ina Vac vs. GP-C2	24.16	-23.82 to 72.13	No	ns	0.7314
Ina Vac vs. GP-C3	-53.46	-101.4 to -5.489	Yes	*	0.0190
Vector vs. WT	-60.27	-108.2 to -12.30	Yes	**	0.0050

Vector vs. WAASR	-54.02	-102.0 to -6.050	Yes	*	0.0171
Vector vs. GP-C1	-46.88	-94.86 to 1.093	No	ns	0.0598
Vector vs. GP-C2	-76.34	-124.3 to -28.37	Yes	***	0.0001
Vector vs. GP-C3	-154.0	-201.9 to -106.0	Yes	****	<0.0001
WT vs. WAASR	6.245	-41.73 to 54.22	No	ns	0.9997
WT vs. GP-C1	13.39	-34.59 to 61.36	No	ns	0.9796
WT vs. GP-C2	-16.07	-64.05 to 31.90	No	ns	0.9498
WT vs. GP-C3	-93.70	-141.7 to -45.72	Yes	****	<0.0001
WAASR vs. GP-C1	7.143	-40.83 to 55.12	No	ns	0.9993
WAASR vs. GP-C2	-22.32	-70.29 to 25.65	No	ns	0.7974
WAASR vs. GP-C3	-99.94	-147.9 to -51.97	Yes	****	<0.0001
GP-C1 vs. GP-C2	-29.46	-77.44 to 18.51	No	ns	0.5153
GP-C1 vs. GP-C3	-107.1	-155.1 to -59.11	Yes	****	<0.0001
GP-C2 vs. GP-C3	-77.62	-125.6 to -29.65	Yes	****	<0.0001
Day 70					
Ina Vac vs. Vector	25.08	-22.90 to 73.05	No	ns	0.6958
Ina Vac vs. WT	-16.20	-64.17 to 31.77	No	ns	0.9480
Ina Vac vs. WAASR	1.274	-46.70 to 49.25	No	ns	>0.9999
Ina Vac vs. GP-C1	-8.494	-56.47 to 39.48	No	ns	0.9982
Ina Vac vs. GP-C2	-14.77	-62.75 to 33.20	No	ns	0.9666
Ina Vac vs. GP-C3	-123.7	-171.7 to -75.70	Yes	****	<0.0001
Vector vs. WT	-41.28	-89.25 to 6.697	No	ns	0.1394
Vector vs. WAASR	-23.80	-71.78 to 24.17	No	ns	0.7447
Vector vs. GP-C1	-33.57	-81.55 to 14.40	No	ns	0.3542
Vector vs. GP-C2	-39.85	-87.82 to 8.126	No	ns	0.1693
Vector vs. GP-C3	-148.8	-196.7 to -100.8	Yes	****	<0.0001
WT vs. WAASR	17.47	-30.50 to 65.45	No	ns	0.9263
WT vs. GP-C1	7.706	-40.27 to 55.68	No	ns	0.9990
WT vs. GP-C2	1.429	-46.55 to 49.40	No	ns	>0.9999
WT vs. GP-C3	-107.5	-155.5 to -59.50	Yes	****	<0.0001
WAASR vs. GP-C1	-9.768	-57.74 to 38.21	No	ns	0.9962
WAASR vs. GP-C2	-16.05	-64.02 to 31.93	No	ns	0.9503
WAASR vs. GP-C3	-125.0	-172.9 to -76.98	Yes	****	<0.0001
GP-C1 vs. GP-C2	-6.277	-54.25 to 41.70	No	ns	0.9997
GP-C1 vs. GP-C3	-115.2	-163.2 to -67.21	Yes	****	<0.0001

GP-C2 vs. GP-C3	-108.9	-156.9 to -60.93	Yes	****	<0.0001
-----------------	--------	------------------	-----	------	---------

Statistical Analysis Example (One-way ANOVA with Tukey's multiple comparisons test):

Fig1m (IL4 ELISpot)

ANOVA summary	
F	12.60
P value	<0.0001
P value summary	****
Significant diff. among means (P < 0.05)?	Yes
R square	0.7297

Tukey's multiple comparisons test	Mean Diff.	95.00% CI of diff.	Significant?	Summary	Adjusted P Value
Ina Vac vs. Vector	299.6	123.0 to 476.2	Yes	***	0.0002
Ina Vac vs. WT	38.00	-138.6 to 214.6	No	ns	0.9926
Ina Vac vs. WAASR	-41.80	-218.4 to 134.8	No	ns	0.9877
Ina Vac vs. GP-C1	147.6	-28.96 to 324.2	No	ns	0.1487
Ina Vac vs. GP-C2	-78.80	-255.4 to 97.76	No	ns	0.7889
Ina Vac vs. GP-C3	-85.40	-262.0 to 91.16	No	ns	0.7226
Vector vs. WT	-261.6	-438.2 to -85.04	Yes	**	0.0011
Vector vs. WAASR	-341.4	-518.0 to -164.8	Yes	****	<0.0001
Vector vs. GP-C1	-152.0	-328.6 to 24.56	No	ns	0.1274
Vector vs. GP-C2	-378.4	-555.0 to -201.8	Yes	****	<0.0001
Vector vs. GP-C3	-385.0	-561.6 to -208.4	Yes	****	<0.0001
WT vs. WAASR	-79.80	-256.4 to 96.76	No	ns	0.7793
WT vs. GP-C1	109.6	-66.96 to 286.2	No	ns	0.4550
WT vs. GP-C2	-116.8	-293.4 to 59.76	No	ns	0.3808
WT vs. GP-C3	-123.4	-300.0 to 53.16	No	ns	0.3186
WAASR vs. GP-C1	189.4	12.84 to 366.0	Yes	*	0.0294
WAASR vs. GP-C2	-37.00	-213.6 to 139.6	No	ns	0.9936
WAASR vs. GP-C3	-43.60	-220.2 to 133.0	No	ns	0.9848
GP-C1 vs. GP-C2	-226.4	-403.0 to -49.84	Yes	**	0.0057
GP-C1 vs. GP-C3	-233.0	-409.6 to -56.44	Yes	**	0.0042
GP-C2 vs. GP-C3	-6.600	-183.2 to 170.0	No	ns	>0.9999

Question 15:

- A healthy control (non-infected) for all tissue histology is missing in the comparison.

Response:

Thank you for highlighting the need for appropriate histological controls. We fully acknowledge the importance of including uninfected controls to establish a baseline for assessing vaccine-mediated protection against tissue injury.

In direct response to this point, we have supplemented all histopathological analyses in the revised manuscript with tissue sections from age-matched, non-infected healthy control mice. These controls are now integrated into the relevant figures (e.g., Fig. 2, Extended Data Fig. 3) and provide the essential reference for interpreting the degree of pathology in challenged animals.

We would also like to provide context regarding the animal model used, as it directly informs the interpretation of histopathology (as detailed in our response to Q11). For Old World hantaviruses like HTNV, an ideal model that fully recapitulates human disease is lacking. The standard murine challenge model employed here is characterized by acute, non-lethal infection with inherently modest histopathological changes, as its primary endpoint is the quantification of peak viral replication. Including uninfected controls is therefore crucial to baseline these subtle, model-specific changes and to rule out procedure-related artifacts.

Furthermore, to ensure transparency and address concerns about field selection, we have adopted a new presentation format for histology images. As detailed in our response to Reviewer #1's Major Point 1, each panel now includes a low-magnification overview alongside higher-magnification views, allowing for assessment of the entire tissue section.

In addition, we have provided below the pathological images at the lowest magnification for each of the above experimental groups for your reference.

Question 16:

- Line 234, figure “1h” is showing IFN γ SFC, you mean “Fig 1g”

Response:

We appreciate your careful review. You are correct. The citation at Line 234 for the IFN γ ELISpot (SFC) results should refer to Fig. 1g, not Fig. 1h. We have corrected this typographical error.

Question 17:

- Line 292, is “4d”

Response:

Thank you for noting this. We have corrected to "4d".

Question 18:

- Line 295, this sentence is not true as it is now well known that one of the weakest points of mRNA vaccines, especially SARS-CoV2 mRNA vaccines is that they lack on effective long term T-cell activation.

Response:

We thank you for raising this important point regarding T cell responses to mRNA vaccines. We acknowledge your perspective and the existing literature, including studies on SARS-CoV-2 mRNA vaccines, which indicate that the induction of potent and durable T cell immunity can be a challenge for some mRNA vaccine platforms.

Upon careful review, we agree that the original phrasing in original line 295 could be misinterpreted as an overgeneralization regarding long-term T cell activation by mRNA vaccines. To address this concern and to precisely reflect our experimental data, we have revised the sentence accordingly. The updated text in the Discussion now reads:

[...the prefusion-stabilized GP-C3 immunogen delivered via mRNA-LNP elicits robust and balanced immune responses, characterized by high-titer, cross-reactive NABs and a Th1-skewed cellular profile, confirming its efficacy across nucleic acid vaccine platforms.]

Question 19:

- One of the interesting parts of this study is the mixed vaccine protocol by priming with the inactivate Ina Vac and boosting with different GP construct. This part, in my opinion, is the most innovative and should've received more focus for the immune response elicited and to suggest a better way to target the immune system to a more specific and protecting HTNV immune response.

Response:

We sincerely thank you for your positive assessment of our heterologous prime-boost strategy and for your valuable suggestion to provide a deeper focus on the immune response it generates. We agree that this approach is particularly innovative, as it aims to steer the immune system toward a more specific and protective response against HTNV, which holds significant promise for populations primed with the inactivated vaccine.

In direct response to your suggestion, we have expanded the discussion of these results in the revised manuscript to elaborate more thoroughly on the potential immunological mechanisms and implications of this strategy.

Moreover, to directly investigate the underlying immunology as your comment inspires, we completed an independent, dedicated animal study exploring this heterologous regimen in depth. This comprehensive study, which involved a lengthy immunization and observation schedule, included B cell receptor (BCR) repertoire sequencing of splenocytes. Preliminary analysis of these data has revealed distinct clonal expansion and repertoire shifts in the heterologous group compared to homologous vaccination controls. For your reference, initial findings suggest a preferential recruitment of B cell clones utilizing specific VH gene segments.

The full mechanistic interpretation of this sequencing data, however, naturally leads to a subsequent and substantial investigation. Determining the neutralizing function, precise mechanisms, and *in vivo* efficacy of the dominant antibody clones requires dedicated work involving antibody isolation, recombinant expression, and comprehensive functional assays. We believe these deep mechanistic insights form a complete and compelling narrative on their own. Therefore, while this proof-of-concept study establishes the foundation, the subsequent mechanistic study is being prepared for submission as an independent manuscript. This approach is further warranted by the need for appropriate intellectual property management concerning the novel antibody sequences identified, which supports presenting this focused work separately.

For your reference, we have included below several key analyses from this sequencing study. The clonality snail plot visually summarizes the dynamics of clonal expansion. The

Circos plot illustrates the correlation and sharing of dominant BCR clonotypes across different experimental groups, providing a network view of the repertoire response. Additionally, the heatmap of IGH V-J gene segment usage highlights the preferential recruitment of B cells with specific genetic signatures in the heterologous regimen. We hope these data underscore the value of this continued line of inquiry. We genuinely appreciate your interest in this direction and hope to have the opportunity to share the full subsequent study with you in the future.

CDR3 of IGH, IGL and IGK length

[Redacted]

IGH V-J gene Circos plot

[Redacted]

Clonality snailplot

[Redacted]

Heatmap of IGH V-J gene

[Redacted]

Point-by-point response to the reviewers' comments

We sincerely thank all reviewers for their time and valuable feedback on our manuscript. We are delighted that the reviewers found the revised version substantially improved and the responses satisfactory. We have carefully considered all comments and have incorporated corresponding revisions into the manuscript, as detailed in the point-by-point responses below. All changes in the manuscript text are highlighted in red for ease of reference.

REVIEWERS' COMMENTS

Reviewer #1 (Remarks to the Author):

The authors have responded very carefully to the reviewers' comments, and the revised manuscript has become a very valuable. I look forward to seeing this vaccine advance toward practical application. In particular, the GP-C3 mRNA vaccine achieved an FRNT50 of approximately 1:250, suggesting that its protective efficacy is likely to be sufficient in vivo.

In this study, post-immunization sera neutralized SEOV; however, it will be important in future work to determine whether cross-neutralization extends to other Old World hantavirus species. In addition, it will be essential to demonstrate that vaccination does not lead to antibody-dependent enhancement upon subsequent infection with heterologous hantaviruses.

Reviewer #3 (Remarks to the Author):

Thanks for the revised manuscript and the detailed replies. The part I cared most about was the potential overlap with your previous HTNV vaccine work and the other closely related paper, and I think you handled that well. Your explanation makes it clear why this is a different story and not just a repeat, especially because the core advance here is the structure guided prefusion stabilization approach and the GP C3 design, and you also make a clear distinction between a true mRNA LNP platform and other delivery setups. What helped is that you also reflected this in the manuscript text, so readers will understand the novelty without having to rely on the rebuttal.

The present revision reads cleaner and the flow is easier to follow. It is now straightforward to see how the DNA experiments were used to identify the lead construct and then how the same lead was tested in the mRNA LNP format. I also appreciate that you acknowledged the B cell level limitation and framed it as a real technical bottleneck rather than ignoring it. At this point, I feel my major concerns are addressed and I do not have additional substantive requests.

RESPONSE TO THE COMMENTS OF REVIEWER

Reviewer #1:

Comment 1

The authors have responded very carefully to the reviewers' comments, and the revised manuscript has become a very valuable. I look forward to seeing this vaccine advance toward practical application. In particular, the GP-C3 mRNA vaccine achieved an FRNT50 of approximately 1:250, suggesting that its protective efficacy is likely to be sufficient in vivo.

Response: Thank you for your positive assessment of our revised manuscript and for your supportive comments regarding the vaccine's translational potential. We sincerely appreciate your recognition of our efforts in addressing the reviewers' feedback.

We also thank you for highlighting these important future research directions regarding broader

cross-neutralization and antibody-dependent enhancement. We agree that these are critical questions for the further development of the vaccine platform, and we will certainly incorporate these considerations into our ongoing and future studies.

Thank you again for your time and valuable input throughout the review process.

Comment 2

In this study, post-immunization sera neutralized SEOV; however, it will be important in future work to determine whether cross-neutralization extends to other Old World hantavirus species.

Response: Thank you for raising this valuable and forward-looking point. We fully agree that defining the breadth of cross-neutralization is a critical next step. In direct response, we have added a dedicated paragraph on future directions in the Discussion section. It explicitly states that determining the cross-protective breadth of GP-C3 against other Old World hantaviruses (e.g., DOBV, PUUV) is a primary objective (See Response to Q3).

Comment 3

In addition, it will be essential to demonstrate that vaccination does not lead to antibody-dependent enhancement upon subsequent infection with heterologous hantaviruses.

Response: We appreciate the reviewer for raising this crucial safety consideration regarding antibody-dependent enhancement (ADE). In direct response, we have integrated this point into the new paragraph in the Discussion. It now states that comprehensive safety evaluations, including specific ADE risk assessment, are a mandatory component of the advanced development pathway. We acknowledge the importance of this issue not only from theoretical grounds but also based on our own prior observations. In a separate cohort study (*Virology*, PMID: 39923054), we identified a convalescent patient serum exhibiting clear *in vitro* ADE activity against HTNV, underscoring its biological plausibility for hantaviruses. Therefore, proactively evaluating ADE risk for our novel prefusion-stabilized immunogen is an essential next step, and we are committed to incorporating specific ADE investigations into the future development plan for the GP-C3 vaccine candidate.

Added paragraph.

Future work will be essential to advance this promising candidate. First, the cross-protective breadth of GP-C3 against other clinically relevant Old World hantaviruses (e.g., DOBV, PUUV) requires empirical determination. Second, and equally critical, comprehensive safety evaluations—including specific assessment for the potential risk of antibody-dependent enhancement (ADE) upon exposure to heterologous hantaviruses—will be a mandatory component of the advanced preclinical and clinical development pathway^{67, 68}. Finally, the prefusion stabilization strategy validated here provides a blueprint for designing novel immunogens against other global threats, including New World hantaviruses such as ANDV.

Ref:

67. Wei J, Zhang H, Pei J, Yang Q, Wang Y, Jin X, et al. Standardization, validation, and comparative evaluation of a convenient surrogate recombinant vesicular stomatitis virus plaque reduction test for quantification of Hantaan orthohantavirus (HTNV) neutralizing antibodies. *Virology journal* 2025, 22(1): 31.

68. Yao JS, Kariwa H, Takashima I, Yoshimatsu K, Arikawa J, Hashimoto N. Antibody-dependent enhancement of hantavirus infection in macrophage cell lines. *Archives of virology* 1992, 122(1-2): 107-118.

Excerpt from our previous publication (*Virology*, PMID: 39923054):

An interesting result was observed with patient NO.11, for whom HTNV infection was enhanced at 1:10 to 1:80 dilutions of serum rather than 1:160. This phenomenon suggests an antibody-dependent enhancement (ADE) effect in this patient, and we conducted further analysis excluding NO.11.

Last, we found an interesting phenomenon where one convalescent serum exhibited an ADE effect at lower dilutions, a phenomenon similar to that observed in respiratory syncytial virus, Dengue virus, and SARS-CoV-2 infection [51–54]. These results may be attributed to non-neutralizing antibody that facilitates virus infection through their Fc domains [55], since one report suggested ADE exists in hantavirus-infected macrophage cell lines [56]. However, the precise mechanism for hantavirus-related ADE still requires further investigation.

Reviewer #3:

Thanks for the revised manuscript and the detailed replies. The part I cared most about was the potential overlap with your previous HTNV vaccine work and the other closely related paper, and I think you handled that well. Your explanation makes it clear why this is a different story and not just a repeat, especially because the core advance here is the structure guided prefusion stabilization approach and the GP C3 design, and you also make a clear distinction between a true mRNA LNP platform and other delivery setups. What helped is that you also reflected this in the manuscript text, so readers will understand the novelty without having to rely on the rebuttal.

The present revision reads cleaner and the flow is easier to follow. It is now straightforward to see how the DNA experiments were used to identify the lead construct and then how the same lead was tested in the mRNA LNP format. I also appreciate that you acknowledged the B cell level limitation and framed it as a real technical bottleneck rather than ignoring it. At this point, I feel my major concerns are addressed and I do not have additional substantive requests.

Response: Thank you so much for your time and for this very positive and encouraging feedback on our revised manuscript.

We are truly grateful that you found our explanations regarding the distinction from prior work to be clear, and that you recognize the core advance of our structure-guided stabilization approach and mRNA LNP platform. Your insightful comments throughout the review process were instrumental in helping us refine the manuscript and sharpen the narrative around its novelty and key findings.

We sincerely appreciate all the time and thought you have dedicated to reviewing our work.